# Temporal dynamics of short-term neural adaptation across human visual cortex

**Amber Marijn Brands** [1]*, **Sasha Devore**[2], **Orrin Devinsky**[2], **Werner Doyle**[2], **Adeen Flinker**[2], **Daniel Friedman**[2], **Patricia Dugan**[2], **Jonathan Winawer** [3], **Iris Isabelle Anna Groen** [1]

**1** Informatics Institute, University of Amsterdam, Amsterdam, The Netherlands, **2** New York University Grossman School of Medicine, New York, New York, United States of America, **3** Department of Psychology, New York University, New York, New York, United States of America

* a.m.brands@uva.nl

**Data Availability Statement:** All code used for the purpose of this paper can be found at the GitHub repositories https://github.com/ABra1993/tAdaptation_ECoG.git and https://github.com/WinawerLab/ECoG_utils. Raw and processed iEEG

## Abstract

Neural responses in visual cortex adapt to prolonged and repeated stimuli. While adaptation occurs across the visual cortex, it is unclear how adaptation patterns and computational mechanisms differ across the visual hierarchy. Here we characterize two signatures of short-term neural adaptation in time-varying intracranial electroencephalography (iEEG) data collected while participants viewed naturalistic image categories varying in duration and repetition interval. Ventral- and lateral-occipitotemporal cortex exhibit slower and prolonged adaptation to single stimuli and slower recovery from adaptation to repeated stimuli compared to V1-V3. For category-selective electrodes, recovery from adaptation is slower for preferred than non-preferred stimuli. To model neural adaptation we augment our delayed divisive normalization (DN) model by scaling the input strength as a function of stimulus category, enabling the model to accurately predict neural responses across multiple image categories. The model fits suggest that differences in adaptation patterns arise from slower normalization dynamics in higher visual areas interacting with differences in input strength resulting from category selectivity. Our results reveal systematic differences in temporal adaptation of neural population responses between lower and higher visual brain areas and show that a single computational model of history-dependent normalization dynamics, fit with area-specific parameters, accounts for these differences.

## Author summary

Neural responses in visual cortex adapt over time, with reduced responses to prolonged and repeated stimuli. Here, we examine how adaptation patterns differ across the visual hierarchy in neural responses recorded from human visual cortex with high temporal and spatial precision. To identify possible neural computations underlying short-term adaptation, we fit the response time courses using a temporal divisive normalization model. The model accurately predicts prolonged and repeated responses in lower and higher visual areas, and reveals differences in temporal adaptation between visual areas and stimulus categories. Our model suggests that differences in adaptation patterns result from

data are publicly available as part of the 'Visual ECoG dataset' on OpenNeuro ( https://openneuro.org/datasets/ds004194). Code used to generate the figures is available at  https://github.com/ABra1993/tAdaptation_ECoG.git.

**Funding:** This work was supported by a MacGillavry Fellowship to IIAG and a NIH R01 MH111417 to JW. The funders had no role in study design, data collection and analysis, decision to publish, or preparation of the manuscript.

**Competing interests:** The authors have declared that no competing interests exist.

differences in divisive normalization dynamics. Our findings shed light on how information is integrated in the brain on a millisecond-time scale and offer an intuitive framework to study the emergence of neural dynamics across brain areas and stimuli.

## Introduction

Neural responses in human visual cortex adapt over time, showing reduced responses to prolonged and repeated stimuli. Adaptation occurs at multiple spatial scales: from single-cell recordings in monkeys [1–3] to neural population responses in humans on functional magnetic resonance imaging (fMRI) [4–7], magneto- and electroencephalography (M/EEG) [8–12] and electrocorticography (ECoG) [13, 14]. Adaptation also occurs at multiple temporal scales, from milliseconds [15] to minutes [16] or days [17]. Adaptation is thought to facilitate efficient neural coding by allowing the brain to dynamically recalibrate to changing sensory inputs [18–20], but its role in visual processing is not precisely understood. For example, it is unclear if adaptation patterns and computational mechanisms differ across visual brain areas.

To elucidate these issues, we studied two signatures of adaptation in time-resolved neural responses at short (sub-second) time-scales. First, neural responses reduce in magnitude when a static stimulus is viewed continuously, evident in transient-sustained dynamics in the shape of response time courses (Fig 1A). Second, when two stimuli are viewed close in time, the response to the second stimulus is reduced; i.e., repetition suppression (RS; [1, 21–23]; Fig 1B). Higher visual areas have been found to show slower transients and more slowly decaying responses than lower visual areas in human ECoG [13, 14] and in simulated neural fMRI responses [24, 25], and fMRI studies suggest that higher visual areas show stronger RS than lower areas (e.g., V1; [7, 26]). Further, a computational model of delayed divisive normalization [27, 28] simultaneously predicts transient-sustained dynamics and RS in neural population responses measured with ECoG [13, 14], implying that both forms of adaptation may reflect divisive normalization mechanisms.

Together, these findings suggest that adaptation signatures differ across the visual hierarchy and that this may reflect differences in history-dependent normalization. However, in most studies, the stimuli were noise patches or simple contrast patterns, which primarily drive

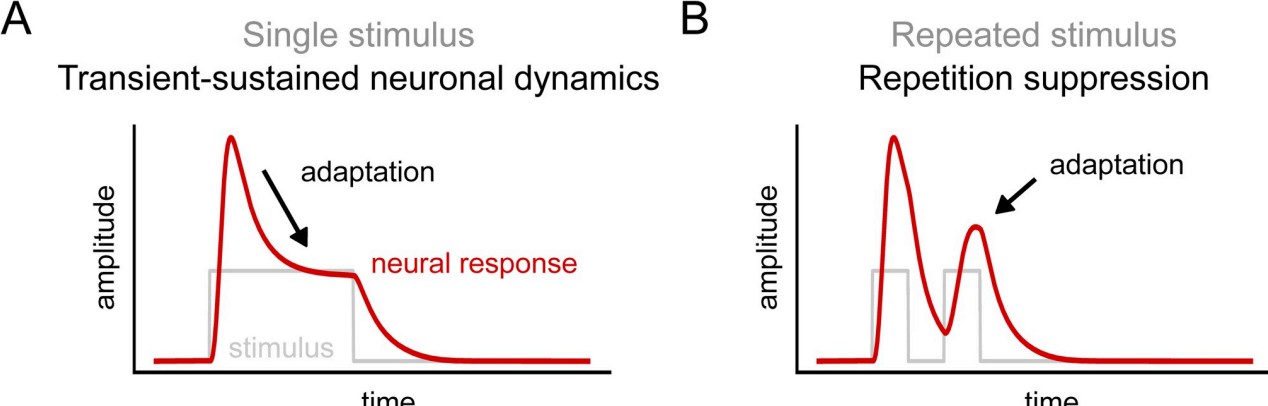

**Fig 1. Two forms of temporal short-term adaptation observed in neural response time courses.** A. For a prolonged single stimulus, adaptation is evident because the neural response, after an initial transient, is followed by a decay plateauing to a sustained response level. B. For two presentations of an identical image with a brief gap in between the stimuli, adaptation is evident because the neural response for the second stimulus is reduced.

responses in lower-level areas. Thus, the observed differences across areas may reflect suboptimal stimuli for higher visual areas, rather than systematic differences in temporal adaptation. Further, neural adaptation also may vary within an area, depending on stimulus type. Monkey and human fMRI studies find that in visual areas with increased sensitivity to stimulus categories such as faces or bodies, preferred stimuli elicit stronger RS than non-preferred stimuli [3, 29, 30]. Thus, to compare and model adaptation across the visual hierarchy, stimulus effectiveness must be considered.

To disentangle the influence of visual area and stimulus on neural adaptation, we quantified transient-sustained and repetition suppression dynamics of neural responses across multiple visual brain regions in a new set of intracranial EEG (iEEG) recordings from human participants. Participants were presented with naturalistic stimuli from distinct image categories, allowing us to assess stimulus preference and its effectiveness on neural adaptation patterns. By fitting an augmented version of the delayed divisive normalization model [13, 14] that considers stimulus category preference, we propose explanations for differences in adaptation patterns.

Our results yield three insights. First, we demonstrate systematic differences in neural adaptation between lower and higher human visual areas: lower areas show faster transient-sustained dynamics and faster recovery from repetition suppression. Second, we reveal stimulus-specific differences in recovery from RS in category-selective electrodes: preferred stimuli elicit stronger repetition suppression than non-preferred stimuli. Third, our augmented DN model accurately predicts neural responses to different stimulus categories along the visual hierarchy. Based on the observed model behavior, we propose that observed differences in neural adaptation patterns reflect differences in divisive normalization dynamics.

## Materials and methods

### Ethics statement

Approval for this study was granted by New York University, Grossman School of Medicine, Institutional Review Board. Prior to the experiment participants gave written informed consent. The methods for collecting and preprocessing the ECoG data have been recently described by [14]. For convenience, the following sections were duplicated with modifications reflecting differences from the previous method: *ECoG recordings*, *Data preprocessing* and *Electrode localization*.

### Subjects

Intracranial EEG data were collected from four participants who were implanted with subdural electrodes for clinical purposes at the New York University Grossman School of Medicine (New York, USA). All participants had normal or corrected-to-normal vision and were implanted with standard clinical strip, grid and depth electrodes. One participant was additionally implanted with a high-density research grid (HDgrid), for which separate consent was obtained. Detailed information about each participant and their implantation is provided in S1 Table and S9 Fig.

### iEEG recordings

Recordings were made using a Neuroworks Quantum Amplifier (Natus Biomedical) recorded at 2048 Hz, band-pass filtered at 0.01–682.67 Hz, and then downsampled to 512 Hz. An audio trigger cable, connecting the laptop and the iEEG amplifier, was used to record stimulus onsets and the iEEG data. Behavioral responses were recorded by an external number pad that was

connected to the laptop through a USB port. Participants initiated the start of the next run by pushing a designated response button on the number pad.

## Stimuli

Stimuli consisted of natural color images presented on a gray background belonging to one of the following six categories: bodies, buildings, faces, objects, scenes and scrambled (Fig 2A). Images (568 x 568 pixels) were taken from a set of stimuli used in prior fMRI studies to localize functional category-selective brain regions [31, 32]. In total the dataset consisted of 288 images with 48 images per category. Bodies consisted of pictures of hands (24 images) and feet (24 images) taken from a variety of viewpoints. Buildings consisted of a large variety of human-built structures (including houses, apartment buildings, arches, barns, mills, towers,

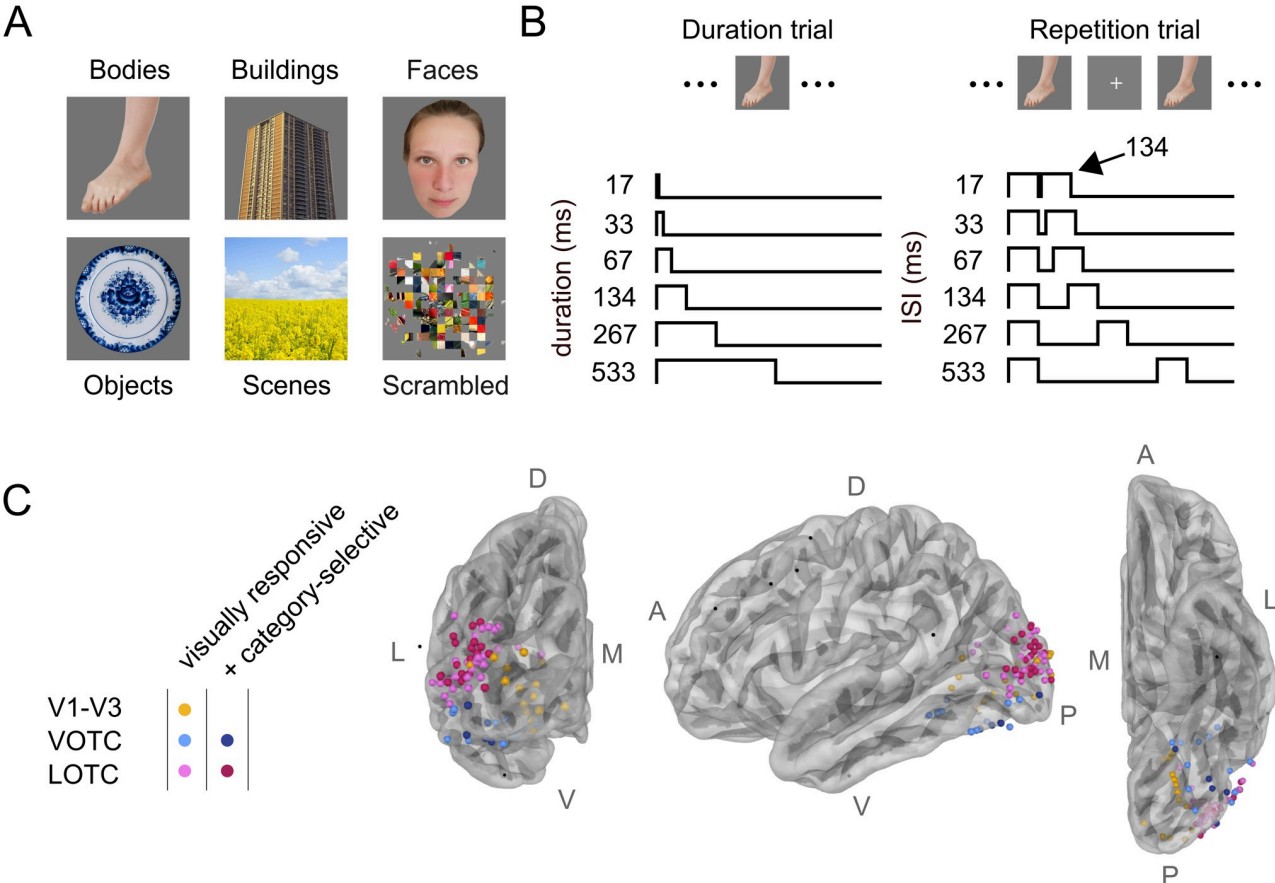

**Fig 2. Experimental design and electrode positions.** A: Stimuli consisted of natural images belonging to one of six image categories (bodies, buildings, faces, objects, scenes and scrambled). For privacy reasons, the face exemplar shown here depicts the first author and for copyright reasons, exemplars from the buildings, objects and scene category are stock photos (source https://www.pexels.com/). These images were not included in the actual dataset. The scrambled exemplars consisted of image patches and were created by taking cropped images and swapping the pixels (mkFigure2_scrambled.m). B: Subjects were presented with two different trial types. Duration trials (left) consisted of a single stimulus with one of six durations, ranging from 17–533 ms. Repetition trials (right) comprised two stimulus presentations of 134 ms each with one of six ISIs ranging from 17–533 ms. Subjects fixated on a small cross and were instructed to press a button whenever it changed color. C: Electrodes with robust visual responses were identified in V1-V3 (n = 15), VOTC (n = 17) or LOTC (n = 47). Electrodes not included in the dataset are shown in black. Electrodes were considered category-selective if the average response for one image category was higher than for the other image categories (d' > 0.5, see Eq 1, Materials and methods, n = 26). Apparent misalignments between electrode positions and the brain surface in C result from the fact that the electrodes here are displayed on the reconstructions of the average brain surface; electrode assignment was performed in each participant's native T1 space (S9 Fig). L = lateral, M = medial, D = dorsal, V = ventral, A = anterior, P = posterior. The brain surfaces were created using MNE-Python and can be reproduced by mkFigure2.py.

skyscrapers, etc). Face images were taken from frontal viewpoints and were balanced for gender (24 male, 24 female) and included a variety in race and hairstyle. Objects consisted of both man-made items (24 images, e.g., household items, vehicles, musical instruments, electronics and clothing) and natural items (24 images, e.g., fruits/vegetables, nuts, rocks, flowers, logs, leaves, and plants). Scene images were equally divided between indoor, outdoor man-made and outdoor natural scenes (16 images each). Faces, bodies, buildings and objects were cropped out and placed on gray-scale backgrounds. Scrambled images consisted of an assembly of square image patches created by taking the cropped object images and randomly swapping 48 × 48 pixel 'blocks' across images and placing them on a gray-scale background. Stimuli were shown on a 15 inch MacBook Pro laptop with a screen resolution of 1280 x 800 pixels (33 cm x 21 cm), which was placed 50 cm from the participant's eyes (at chest level), resulting in stimuli subtending 8.5 degrees of visual angle. Stimuli were presented at a frame rate of 60 Hz using Psychtoolbox-3 [33–35].

## Experimental procedure

Participants viewed two different types of trials (Fig 2B). Duration trials showed a single stimulus for one of six durations, defined as powers of two times the monitor dwell time (1/60): 17, 33, 67, 134, 267 and 533 ms. Repetition trials contained a repeated presentation of the same image with fixed duration (134 ms) but variable inter-stimulus interval (ISI), ranging between 17–533 ms (same temporal step sizes as the duration trials). These temporal parameters were identical to previous studies [7, 13, 14], but here naturalistic color images were presented instead of gray-scale noise patterns. Each participant underwent 2–6 runs of 144 trials each, including 72 duration trials and 72 repetition trials, which each contained 12 stimuli from each of the six stimulus categories. Trial order was randomized, with an inter-trial-interval (ITI) randomly chosen from a uniform distribution between 1.25–1.75s. Participants were instructed to fixate on a cross at the center of the screen and press a button when it changed from black to white or vice versa. Fixation cross changes occurred independently of the stimulus sequence on randomly chosen intervals between 1–5 s. In between runs participants were allowed a short break. Stimuli were divided into two sets, one for even and one for odd runs, with each set containing 72 of the 144 stimuli. The number of odd/even run pairs determined the number of repetitions for a specific trial-type. Detailed information about the amount of data collected for each participant is provided in S1 Table. Three participants (p12–14) additionally viewed repetition trials in which the second image differed from the first (either a different exemplar from the same category or a different category). These trials are included in the dataset (see Data Availability) but not further analyzed for the purpose of this study.

## iEEG data analysis

**Data preprocessing**   Data was read into MATLAB 2020b using the Fieldtrip Toolbox [36] and preprocessed with custom scripts available at https://github.com/WinawerLab/ECoG_utils. The raw voltage time series from each electrode, obtained during each recording session, were inspected for spiking, drifts or other artifacts. Electrodes were excluded from analysis if the signal exhibited artifacts or epileptic activity, determined based on visual inspection of the raw data traces and spectral profiles, or at the clinician's indication. Next, data were divided into individual runs and formatted according to the iEEG-BIDS format [37]. For each run, the data were re-referenced to the common average computed separately for each electrode group (e.g. grid or strip electrodes, see bidsEcogRereference.m) and a time-varying broadband signal was computed for each run (see bidsEcogBroadband.m): First, the voltage-traces were band-pass filtered by applying a Butterworth filter (passband ripples < 3 dB, stopband attenuation

60 dB) for 10 Hz-wide bands ranging between 50–200 Hz. Bands that included frequencies expected to carry external noise were excluded (60, 120 and 180 Hz). Next, the power envelope of each band-pass filtered time course was calculated as the square of the squared magnitude of the analytic signal. The resulting envelopes were then averaged across bands using the geometric mean (see ecog_extractBroadband.m), ensuring that the resulting average is not biased towards the lower frequencies. The re-referenced voltage and broadband traces for each run were written to BIDS derivatives directories.

**Electrode localization**    Pre- and post-implantation structural MRI images were used to localize intracranial electrode arrays [38]. Electrode coordinates were computed in native T1 space and visualized onto pial surface reconstructions of the T1 scans, generated using FreeSurfer [39]. Boundaries of visual maps were generated for each individual participant based on the preoperative anatomical MRI scan by aligning the surface topology with two atlases of retinotopic organization: an anatomically-defined atlas [40, 41] and a probabilistic atlas derived from retinotopic fMRI mapping [42] (S9 Fig). Using the alignment of the participant's cortical surface to the fsaverage subject retrieved from FreeSurfer, atlas labels defined on the fsaverage were interpolated onto the cortical surface via nearest neighbor interpolation. Electrodes were then matched to both the anatomical and the probabilistic atlas using the following procedure (bidsEcogMatchElectrodesToAtlas.m): For each electrode, the distance to all the nodes in the FreeSurfer pial surface mesh was calculated and the node with the smallest distance was determined to be the matching node. The matching node was then used to assign the electrode to one of the following visual areas in the anatomical atlas (hereafter referred to as the Benson atlas): V1, V2, V3, hV4, VO1, VO2, LO1, LO2, TO1, TO2, V3a, V3b, or none; and to assign it a probability of belonging to each of the following visual areas in the probabilistic atlas (hereafter referred to as the Wang atlas): V1v, V1d, V2v, V2d, V3v, V3d, hV4, VO1, VO2, PHC1, PHC2, TO2, TO1, LO2, LO1, V3b, V3a, IPS0, IPS1, IPS2, IPS3, IPS4, IPS5, SPL1, FEF, or none. After localization, all electrodes were assigned to one of three visual electrode groups: early (V1-V3), ventral-occipital (VOTC) and lateral-occipital (LOTC), according to the following rules (S2 Table): electrodes were assigned to V1-V3 if located in V1, V2, V3 according the Benson atlas or if located in V1v, V1d, V2v, V2d, V3v, V3d according to the Wang atlas. Electrodes were assigned to VOTC if located in hV4, VO1 VO2 according to either the Benson or Wang atlas. Electrodes were assigned to LOTC if electrodes were located in any of the remaining retinotopic atlas areas (with exception of SPL1 and FEF). Electrodes that showed robust visual responses according to the inclusion criteria (see Data selection) but were not matched to any retinotopic atlas region (i.e. that obtained the label 'none' from the retinotopic atlas matching procedure described above), were manually assigned to one of the three groups based on visual inspection of their anatomical location and proximity to already-assigned electrodes (e.g. being located on the same electrode strip extending across the lateral-occipital surface, or penetrating the same cortical region as nearby depth electrodes being assigned to V1-V3). Detailed information about the subject-wise electrode assignment is provided in S3 Table. A schematic layout of the electrodes assigned to the visual regions pooled across all four participants is shown in Fig 2C.

**Data selection**    Python scripts used for data selection can be found at https://github.com/ABra1993/tAdaptation_ECoG.git. Two consecutive data selection steps were performed: 1) trial selection and 2) selection of visually-responsive and category-selective electrodes.

**Trial selection**. Trial selection was performed on the broadband time courses for each electrode separately (analysis_selectEpochs.py). We first computed the maximum (peak) response within each trial, after which the standard deviation (SD) of these maximum values over all trials was computed. Trials were excluded from analysis if the maximum

response was $> 2$ SD. Across participants, on average 3.25% of epochs (min: 1.81%, max: 3.80%) were rejected. Next, broadband time courses were converted to percentage signal change by point-wise dividing and subtracting the average prestimulus baseline (100 to 0 ms prior to stimulus onset) across all epochs within each run (analysis_baselineCorrection. py).

**Electrode selection**. Electrode selection was performed separately for i) the analyses focusing on comparison of temporal dynamics across visual areas and ii) the analyses focusing on comparison across stimuli in category-selective regions.

*Selection of visually-responsive electrodes.* For the comparisons across areas, electrodes were included when showing a robust broadband response based on the following two metrics computed onto the duration trials (analysis_selectElectrodes.py): the z-score ($z - score = \frac{\mu}{\sigma}$) where the mean and deviation are computed across time samples and the onset latency of the response computed over the average stimulus duration. The onset latency was determined during the 150 time samples ($\sim 300$ ms) time window after stimulus onset. First, responses were z-scored, after which the onset latency was defined as the first time point at which the response passed a threshold (0.85 std) for a duration of at least 60 time samples ($\sim 120$ ms). Note, converting responses to z-score was only applied during the electrode selection procedure. The reason for this is because response magnitudes when expressed as a percent signal change vary highly across electrodes. To determine the response onset latency, we were interested in the relative increase after presenting the stimulus, and for this reason time courses were converted to a z-score. Electrodes were included in the final selection when i) a response onset could be determined and ii) if the z-score $> 0.2$. Based on the selection methods described above, on average 37% (min: 14%, max: 57%) of the electrodes assigned to a visual group either according to the Benson or Wang atlas were included.

*Selection of category-selective electrodes.* Electrodes were considered category-selective if they preferentially responded to a given image category over other image categories (excluding scrambled) computed for the duration trials. Category-selectivity of an electrode was measured as $d'$:

$$d' = \frac{\overline{X}_{cat} - \overline{X}_{other}}{\sqrt{\frac{\sigma^2_{cat} + \sigma^2_{other}}{2}}} \tag{1}$$

where $X_{cat}$ and $\sigma_{cat}$ represent the mean response and standard deviation for one image category over time, while $X_{other}$ and $\sigma_{other}$ represent the mean response and standard deviation over time for the other image categories. Category-selective electrodes generally exhibit a low z-score for the non-preferred image categories, possibly leading to exclusion from analysis when considering only the z-score computed over all categories (see above). Therefore, for the comparison across stimuli in category-selective regions, electrodes were included if i) a onset latency for the averaged response over all categories for the duration trials was present and if ii) $d'$ passed a threshold of 0.5, 0.75 or 1. The reason for using a range of threshold values was to verify whether the observed data patterns depend on the chosen threshold, whereby a lower threshold allows inclusion of electrodes which show weaker selectivity for a specific image category (analysis_selectElectrodes.py). Detailed information about the number of category-selective electrodes included is provided in S4 Table and a schematic layout of the category-selective electrodes is shown in Fig 2C for a $d'$ threshold of 0.5 (see S10(A) and S10(B) Fig for a threshold of 0.75 and 1.0, respectively).

**Data summary** The data preprocessing, electrode localization and data selection procedures outlined above resulted in 79 electrodes with robust visual responses over either V1-V3 (n = 17), VOTC (n = 15) or LOTC (n = 47). A subset of these electrodes showed selectivity for specific image categories where the number of category-selective electrodes depended on the threshold of $d'$ (n = 26, n = 12, n = 6 for a threshold of 0.5, 0.75 and 1 respectively). After averaging the time series within trial types, there were 72 response time courses per electrode: 12 temporal conditions (6 durations and 6 ISIs) times 6 image categories. The time series from these 72 conditions were used to investigate the temporal profile of neural adaptation and constituted the data for model fitting.

## Computational modelling

**Model fitting** Computational models and associated model fitting procedures were implemented using custom Python code available at https://github.com/ABra1993/tAdaptation_ECoG.git. Models were fitted separately to individual electrodes, after which parameters or metrics derived from these fits were averaged within visual areas using a bootstrapping procedure described below. Models were fitted using a nonlinear least-squares algorithm (scipy.optimize.least_squareas, SciPy, Python), with bounds on the parameters. The starting points, and upper and lower bounds that were used for fitting can be found at modelling_utils_paramInit.py.

**Models**

**Delayed normalization model**

The broadband time courses for each individual electrode were fitted with a delayed divisive normalization (DN) model previously described conceptually in the appendices of [27, 28] and implemented in [13] and [14]. In the DN model, an input drive is divisively normalized by its own delayed activation history, implemented as a low-pass filter on the input drive (DN.py). The model takes a stimulus time course as input and produces a predicted neural response time course as output, by applying a series of transformations which take the form of a Linear-Nonlinear-Gain control (LNG) structure, corresponding to filtering (L), exponentiation (N), and normalization (G). The model contains four free parameters of interest: $\tau_1$, $n$, $\sigma$ and $\tau_2$ (Fig 3A). In addition, two nuisance parameters are fitted, including a shift (delay in response onset relative to stimulus onset) and electrode-specific scale (i.e. gain of response) to take into account differences in overall response latency and amplitude between electrodes. In the following, we will drop the time index for brevity, and denote free parameters between parentheses.

The input drive, $r_{input\ drive}$, is computed by first convolving a stimulus time course ($s = 0$ when stimulus is absent, $s = 1$ when the stimulus is present) with an impulse response function (IRF), $h_1(\tau_1)$, yielding a linear response prediction:

$$r_L = s * h_1(\tau_1) \tag{2}$$

where $h_1$ is defined as:

$$h_1(\tau_1) = te^{-t/\tau_1} \tag{3}$$

The parameter $\tau_1$ is a time constant and determines the peak (i.e. function peaks when $t = \tau_1$). The input drive is obtained by converting the linear response to a nonlinear response by applying a full-wave rectification and an exponentiation with $n$:

$$r_{input\ drive} = |r_L|^n \tag{4}$$

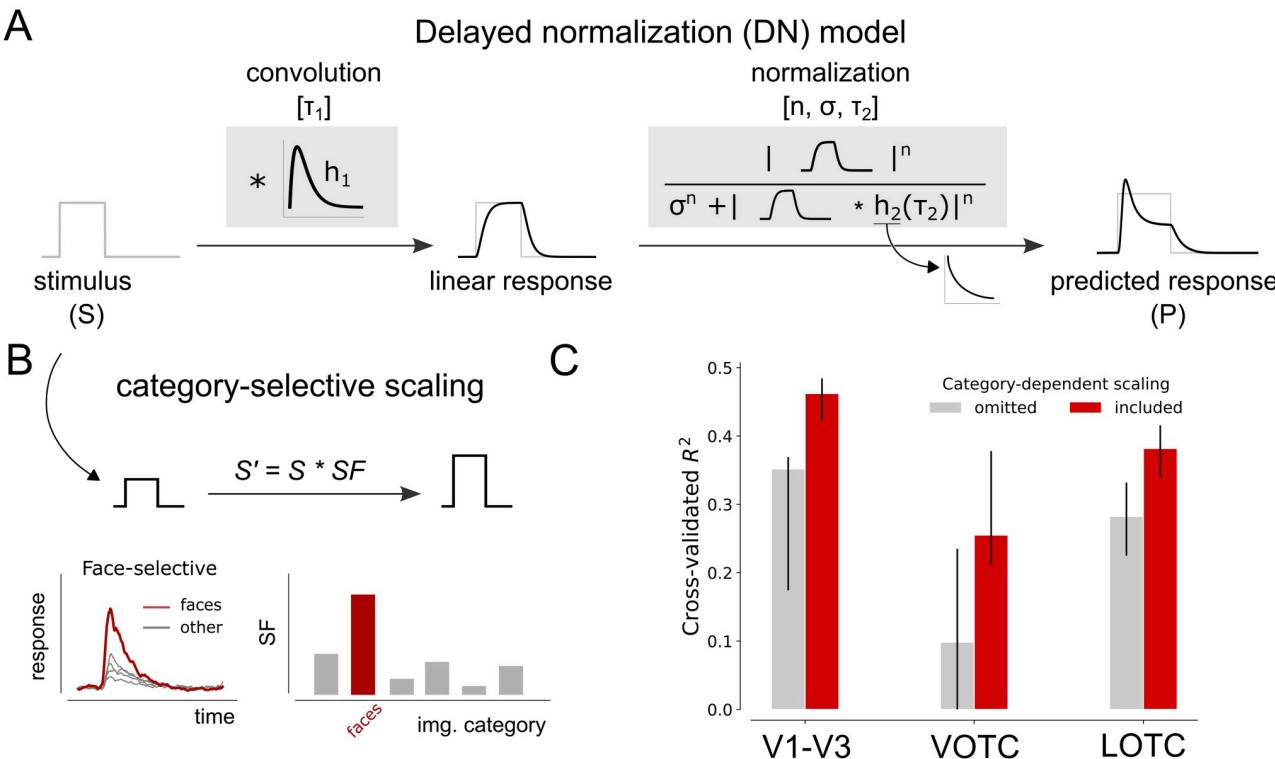

**Fig 3. Modeling neural responses using delayed divisive normalization with category-dependent input strength.** A: Schematic depiction of the delayed divisive normalization (DN) model, originally proposed by [27, 28] and first presented in this form in [13]. The model is defined by a linear-nonlinear-gain control structure, taking as input a stimulus time course and producing a predicted neural response as output. The linear computation consists of a convolution with an impulse response function (IRF), $h_1$, parameterized as a gamma function with $\tau_1$ as a free parameter. The nonlinear computation consists of rectification, exponentiation with a free parameter, $n$, and division by a semi-saturation constant $\sigma$, which is summed with a delayed copy of the input that is also rectified and exponentiated. The delay is implemented as a convolution of the linear response with an exponentially decaying function, $h_2$, with a time constant, $\tau_2$. B: Top, To capture category-selectivity in neural responses, a scaling factor (SF) for each image category is introduced to the DN model, which allows the input drive to vary depending on stimulus category by scaling either the height of the stimulus or predicted neural time course. Bottom, Category-wise responses for an example electrode showing increased sensitivity for faces (left). The category-selectivity is captured by the csDN model evident by the increased value for the face compared to the other image categories. C: Cross-validated explained variance (coefficient of determination) across all stimulus conditions for the DN (omitting category scaling) and the augmented DN model (including category-scaling) plotted per visual area (V1-V3, VOTC and LOTC). The DN model which includes category-specific scaling of the stimulus time course better predicts neural responses across all visual areas. Averages indicate medians and error bars indicate 68% confidence interval across 1000 samples derived from the bootstrapped $R^2$ values. Panel C can be reproduced by mkFigure3.py.

The normalization pool, $r_{normalization}$, is computed by summing a semi-saturation constant, $\sigma$, and a convolution of the linear response with a low-pass filter followed by rectification, where both terms are exponentiated with $n$:

$$r_{normalization}(\sigma, n) = \sigma^n + |r_L * h_2(\tau_2)|^n \tag{5}$$

with the low-pass filter taking the form of the following decaying exponential function with a time constant $\tau_2$:

$$h_2(\tau_2) = e^{-t/\tau_2} \tag{6}$$

In summary, the delayed divisive normalization is applied as follows:

$$r_{DN}(\sigma, n) = \frac{r_{input\ drive}}{r_{normalization}} = \frac{|r_L|^n}{\sigma^n + |r_L * h_2(\tau_2)|^n} \tag{7}$$

The computation of the temporal dynamics by the DN model as described in Eq 7 has the form of a canonical divisive normalization [43], where the normalization pool (i.e. the denominator) consists of a delayed version of the numerator, yielding an output that is characterized by a transient response rise followed by a decay to a sustained response level.

**Augmented DN model with category-selective stimulus strength**

The DN model, as described in the previous section, only receives information about the presence or absence of a stimulus over time. While previous studies [13, 14] scaled the stimulus time course to represent variations in stimulus contrast, they did not incorporate information about the content of a stimulus, e.g., the category it belongs to. Here, we incorporated stimulus content into the model by including six additional free parameters, i.e. one for each image category, which adjust the height of the input stimulus time course according to category preference (Fig 3B), referred to as categorical scaling factors (csDN. py). More specifically, the scaled stimulus course is computed by multiplying the original time course with the scaling factor for the respective category, $s = s \times sf$ where $sf \in sf_{bodies}$, $sf_{buildings}$, $sf_{faces}$, $sf_{objects}$, $sf_{scenes}$, $sf_{scrambled}$ (e.g. for a face stimulus, $s' = s * sf_{face}$). From here on out, we will refer to this augmented version of the DN model as the category-selective DN (csDN) model.

**Model evaluation**   Model performance was quantified as the cross-validated coefficient of determination (modelling_modelFit.py). A 12-fold cross-validation was performed on 72 input broadband time courses, whereby parameters were fitted on 66 conditions and testing was done on the remaining 6 conditions. Within each fold, test data were selected in a pseudo-random manner whereby each image category was always present in one of the six test samples (but these were drawn randomly from the 12 temporal conditions). Comparison of model performance between the DN model which either omits or includes category-dependent scaling confirms that scaling the stimulus course improves model accuracy in all visual areas (Fig 3C). Due to the fact that model accuracy is computed on the left-out data, this result is not guaranteed simply due to adding more free parameters to the DN model. Model parameter values and summary parameters were estimated based on a separate fit to the full dataset.

Note that for electrodes assigned to VOTC the cross-validated explained variance is lower compared to the other visual groups when considering the DN model which omits category scaling. This is likely due to the fact that a large proportion of the electrodes in VOTC show strong category-selectivity, which results in poor model fits for image categories which elicit weaker responses (S11 Fig).

## Summary metrics

To quantify adaptation and associated temporal dynamics, we computed the following summary metrics from the neural time courses and the model predicted time courses:

**Time-to-peak**. The time interval between stimulus onset and the maximum (peak) of the response time course. This metric was computed based on response or model predictions for the longest stimulus duration of 533 ms.

**Full-width at half maximum**. The time from when the response has risen to half of its maximum until it has decayed to half of its maximum. This metric was computed on the neural response or model prediction for all stimulus durations separately.

**Recovery from adaptation for repeated stimuli**. The response magnitude of the second stimulus divided by the first. To obtain a robust estimate of the response to the first stimulus, we averaged together the response time courses for the 134 ms duration stimulus (same duration as the ISI stimuli) and each of the ISI stimuli from trial onset up to the onset of the second stimulus. We then subtracted this average time course from each of the ISI varying stimulus responses, yielding an estimate of the response of the second stimulus corrected for the response to the first stimulus (S3(A) Fig, see also [14]). Subsequently, the recovery from adaptation is defined as the AUC for the second response proportional to the AUC of the first response.

**Overall adaptation for repeated stimuli**. Overall adaptation was computed as the recovery (see above) averaged over all ISIs.

**Long-term recovery**. The amount of recovery (see above) predicted for an ISI of 1 second. To compute this value, a log curve was fitted through recovery values over all ISIs (S3(B) Fig):

$$y = c + a \cdot log(x) \tag{8}$$

where $y$ is the recovery, $x$ is the ISI and $[c, a]$ are two free parameters. When the ISI is 1 second, $y = c$ (because $log(1) = 0$), so $c$ quantifies long-term recovery from neural adaptation. Note that this function is a heuristic applied to short to medium time scales. For very long ISIs, it will make unreasonably large predictions.

## Bootstrapping procedure and statistical testing

When computing the summary metrics outlined above, we repeatedly ($n$ bootstraps) sampled $k$ electrodes with replacement and calculated the mean, followed by computing the summary metric over the averaged timecourse. The median and 68% confidence interval were then computed over the samples derived from the bootstrapped timecourses. Statistical significance was determined by a two-tailed sign test (statistical significance, $\alpha = 0.025$), whereby the difference between two bootstrap distributions was computed and the minimal amount of instances where differences were either positive or negative were divided by $n$. We applied a Bonferroni correction for the number of pairwise comparisons made in the analyses comparing different visual areas (i.e. statistical significance, $\alpha = 0.025/3 = 0.008$).

## Results

We collected iEEG recordings while participants viewed single and repeated naturalistic images from six stimulus categories (Fig 2A), with variable stimulus duration and inter-stimulus-intervals (ISI) (Fig 2B). By aggregating responses across four patients, we identified 79 visually responsive electrodes which we separated into one lower-level visual group (V1-V3) and two higher-level ventral-occipital cortex (VOTC) and lateral-occipital cortex (LOTC) groups using retinotopic atlases (Fig 2C). Some electrodes in VOTC and LOTC were category-selective, showing higher sensitivity to one stimulus class (Fig 2C; see Materials and methods, Data selection). We computed a single average time-resolved broadband response for each temporal stimulus condition and stimulus class, resulting in 72 response time-courses per electrode.

To model neural response dynamics across visual areas and stimuli, the time courses were fitted using a delayed divisive normalization (DN) model. The model takes as input a stimulus time course and produces as output a predicted neural response (Fig 3A). To take into account category-selectivity, we allowed the model to scale the input stimulus time course as a function of category (Fig 3B, see Materials and methods, Computational modeling). Incorporating category-dependent scaling improves model predictions in all visual areas (Fig 3C).

We compared the DN model to temporal two-channel models [24, 44], which we augmented such that it similarly employs a scaling factor for modulating category-specific input strength. These models predict neural responses using distinct channels responsible for either the transient and sustained responses observed in neural signals [45] and has been shown to accurately predict some aspects of iEEG responses [14] and fMRI responses [7, 25]. We distinguished two different model implementations. The L+Q model [24] consists of a linear sustained channel and a transient channel with quadratic nonlinearity, whereas the A+S model [44] contains a sustained channel with adaptation and a transient channel with sigmoid nonlinearities. In the current data, the DN model outperforms the L+Q model in V1-V3 and LOTC and the A+S model in V1-V3 (S1(A) Fig). While for the higher visual regions the A+S model [44] performs nearly on par with the DN model, we see a qualitatively poorer fits with the data, which we will discuss in more detail below.

In the following sections, we first characterize transient-sustained dynamics and repetition suppression in lower and higher visual areas and then examine repetition suppression within category-selective electrodes. Along with the neural data, we present DN model predictions to demonstrate how the observed differences result from divisive normalization dynamics.

## Higher visual areas exhibit slower and prolonged responses to single stimuli

Neural time courses to duration-varying stimuli in V1-V3, LOTC and VOTC exhibit different transient-sustained dynamics (Fig 4A, top panel). In all areas, responses show an initial transient, which for short durations is the only part of the response, while for longer durations, a subsequent lower-amplitude sustained response emerges. However, electrodes in V1-V3 show faster and shorter transients with relatively low sustained responses, while VOTC and LOTC have slower and wider transients with higher sustained responses.

To quantify these differences in response shapes across visual areas, we computed two metrics which capture different characteristics of the time courses. First, responses rise more slowly in higher visual areas as reflected by the time-to-peak, which is shortest for V1-V3, intermediate for VOTC and longest for LOTC (Fig 4B, circle markers). Second, compared to V1-V3, responses for VOTC and LOTC show a broadening of the transient as reflected by the full-width at half-maximum (Fig 4C, circle markers). This difference becomes more pronounced as the stimulus duration lengthens for LOTC and to a lesser degree for VOTC. These metrics indicate a slower rise and a slower decay of the response, resulting in a prolonged, more slowly adapting response in higher visual areas.

The DN model accurately captures the broadband responses for the duration trials across all visual areas (Fig 4A, lower panel). It also predicts the differences in response shapes, that is the slower rise (Fig 4B, triangle markers) and the wider transients (Fig 4C, triangle markers) in LOTC compared to V1-V3, with intermediate values for VOTC. While the A+S model captures the overall response shapes for the different visual areas, it also shows some model failures. First, the model predicts offset responses for longer stimulus durations which are not present in the neural data (S1(B) Fig). Moreover, model predictions show narrower response widths for electrodes in VOTC more similar to those observed in V1-V3 (S1(C) Fig).

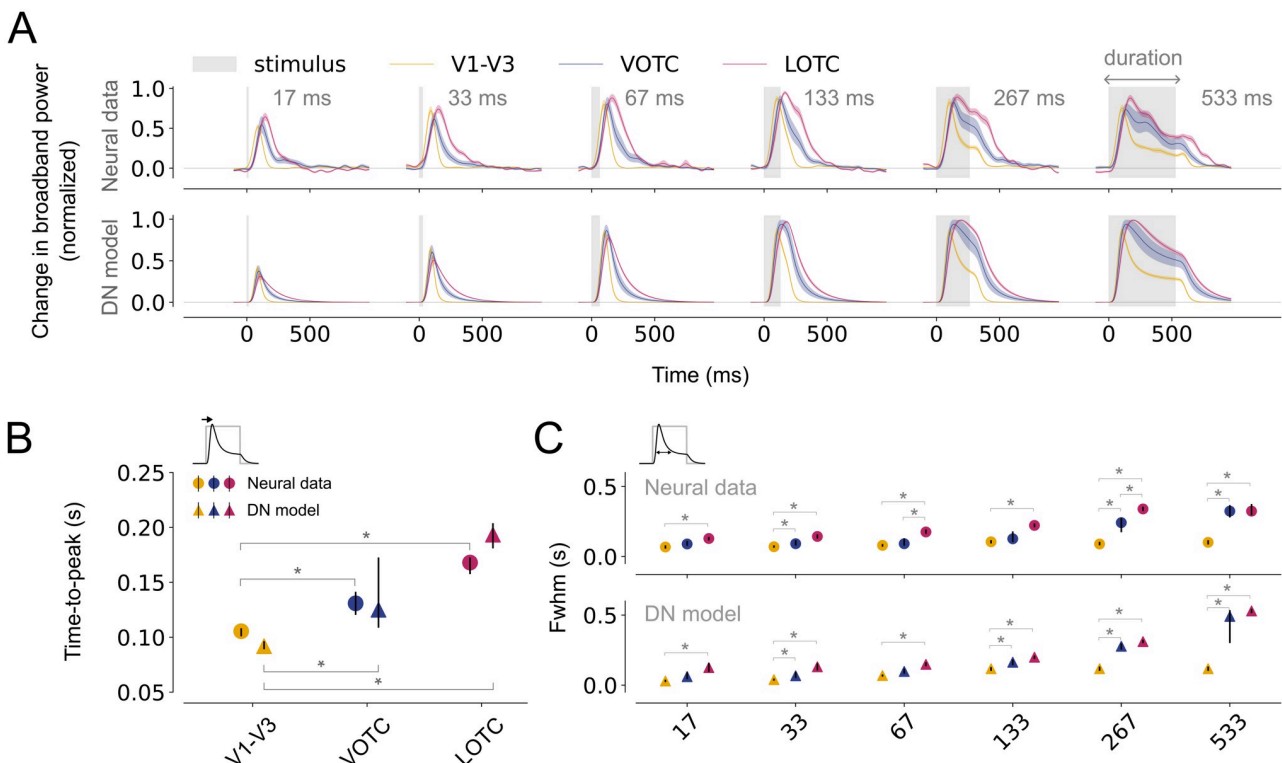

**Fig 4. Slower rise and prolonged responses in higher visual areas.** A: Top, Average, normalized broadband iEEG responses (80–200 Hz) for electrodes assigned to V1-V3 (n = 17), VOTC (n = 15) and LOTC (n = 47) to single stimuli (gray). Responses are shown separately per duration from shortest (17 ms, left) to longest (533 ms, right). Bottom, DN model predictions for the same conditions. The shapes of the neural time courses differ between visual areas and are accurately captured by the DN model. Time courses were smoothed with a Gaussian kernel with standard deviation of $\sigma =$ 10; the shaded regions indicate 68% confidence interval across 1000 bootstrapped timecourses (see Materials and methods, Bootstrapping procedure and statistical testing). B-C: Summary metrics plotted per visual area derived from the neural responses (circle marker) or model time courses (triangle marker). Time-to-peak (B) computed to the longest duration (533 ms). Full-width at half maximum (C), computed for each stimulus duration. Data points indicate medians and error bars indicate 68% confidence interval across 1000 samples derived from the bootstrapped timecourses. Bootstrap test, * = $p < 0.05$ (two-tailed, Bonferroni-corrected). This figure can be reproduced by mkFigure4.py.

To assess whether the differences in transient-sustained dynamics across areas are affected by stimulus selectivity, we quantified these dynamics separately for each electrode's preferred category (eliciting the maximum response) and for all remaining stimulus categories combined (non-preferred stimuli). While for higher visual areas, the response decay for preferred stimuli seems to be slightly stronger compared to non-preferred stimuli, the neural and DN model time courses overall exhibit the same area-specific differences regardless of whether preferred (S2(A)–S2(C) Fig) or non-preferred (S2(D)–S2(F) Fig) stimuli were shown. This suggests that the transient-sustained dynamics in higher visual regions are not stimulus-dependent, but rather reflect intrinsically slower temporal integration.

## Stronger RS and a slower recovery in higher visual areas for repeated stimuli

Viewing repeated stimuli results in repetition suppression in all visual areas (Fig 5A, top panel), whereby responses to the second stimulus are most suppressed at shortest ISIs and show a gradual recovery as ISI increases. Across conditions, there also appear to be differences in RS between lower and higher visual areas. However, quantifying differences in the degree of recovery in these response time courses is not straightforward: the response to the first

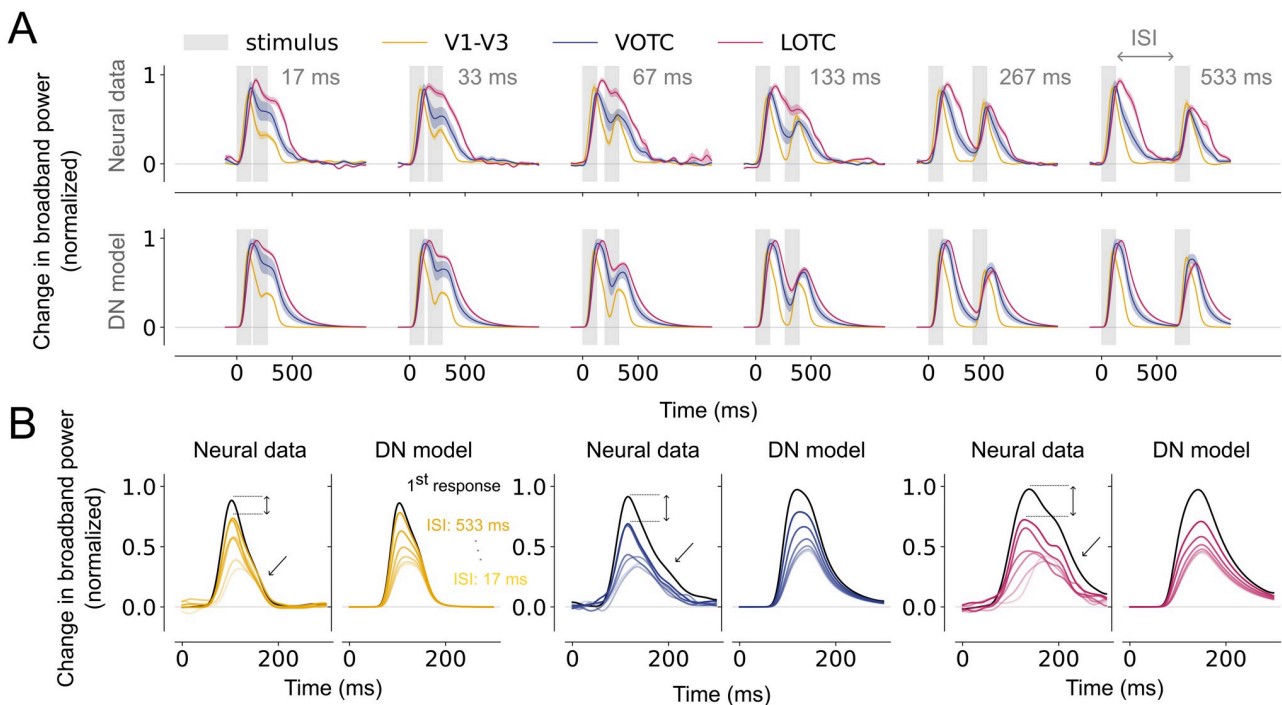

**Fig 5. Qualitative differences in repetition suppression across visual areas.** A: Top, Average, normalized broadband responses for electrodes assigned to V1-V3 (n = 17), VOTC (n = 15) and LOTC (n = 47) to repeated visual stimuli (gray). Responses are shown separately per ISI from shortest (17 ms, left) to longest (533 ms, right). Bottom, DN model predictions for the same data. Time courses differ between visual areas which is captured by the DN model. B. Estimated, normalized response to the second stimulus for V1-V3, VOTC and LOTC. For each visual area, the left panel shows the neural data and the right panel shows the model prediction. Recovery from adaptation gradually increases as the ISI becomes longer, and the rate of recovery is higher for V1-V3 compared to VOTC and LOTC in both the neural data and the DN model as a result of a higher peak magnitude to the second stimulus and a less strong decay after the peak (black arrows). This figure can be reproduced by mkFigure5_6.py.

stimulus continues after its offset (see Fig 4A), and as demonstrated above, this continued response is longer in higher visual regions (Fig 4C). This problem is especially evident for short ISIs: at 17 ms ISI, response amplitudes measured after onset of the second stimulus are higher in LOTC and VOTC than in V1-V3 (Fig 5A), but this could result from weaker RS of the second stimulus, the continued neural responses to the first stimulus, or a combination.

To disentangle these responses, we estimated the response to the second stimulus in isolation (see Materials and methods, Summary metrics) while correcting for the ongoing activity caused by the first stimulus (Fig 5B, Neural data). This shows that recovery from RS qualitatively differs between visual areas: V1-V3 shows less suppression and recovers faster than VOTC and LOTC. These differences between areas are partly due to differences in the peak magnitude of the response to the second stimulus, as well as the faster decay after the peak for higher visual areas (Fig 5B). We quantified the level of RS in these responses by computing their Area Under the Curve (AUC) divided by the AUC of the first stimulus response (see S3 Fig). Neural responses show overall stronger RS for shorter compared to longer ISIs (Fig 6A, left), but also relatively more RS in VOTC and LOTC than in V1-V3. Responses in V1-V3 are nearly fully recovered at the longest ISI of 533 ms, while VOTC and LOTC are still suppressed. Summary metrics of the average recovery across ISIs (Fig 6B, circle markers) and long-term recovery (Fig 6C, circle markers) confirm that there is less RS and faster recovery in V1-V3 compared to VOTC and LOTC.

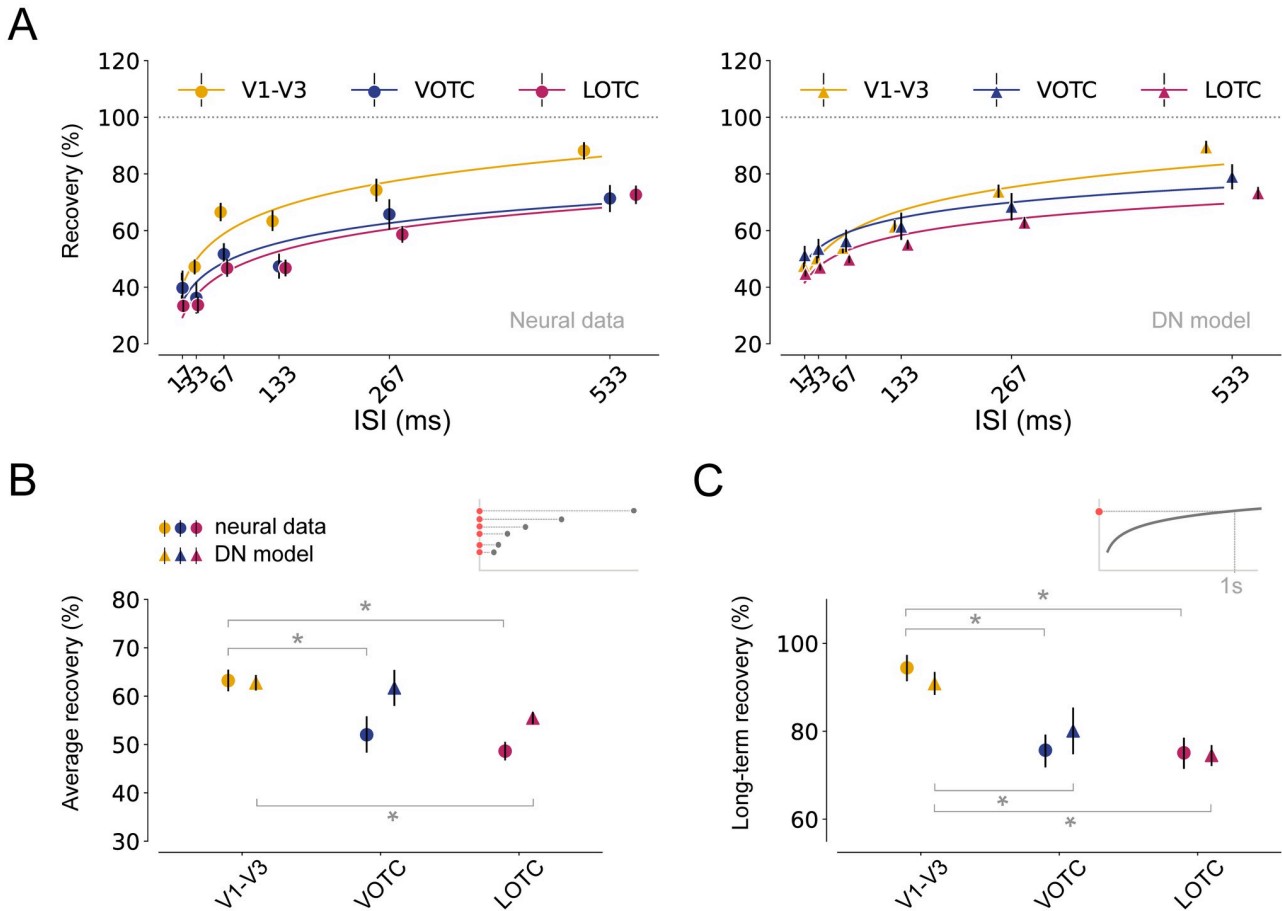

**Fig 6. Stronger RS and slower recovery rate from adaptation in higher visual areas.** A: left, Recovery from adaptation for V1-V3 (n = 17), VOTC (n = 15) and LOTC (n = 47), computed as the ratio of the Area Under the Curve (AUC) between the first and second response. The fitted curves express recovery as a function of the ISI (see Materials and methods, Summary metrics). Higher visual areas show stronger RS and slower recovery from adaptation. Right, model predictions for the same data. The model captures area-specific recovery from adaptation. B-C: Summary metrics plotted per visual area derived from the neural responses (circle markers) or model time courses (triangle markers). Average recovery (B) from adaptation for each area, computed by averaging the AUC ratios between the first and second stimulus over all ISIs. The long-term recovery (C) reflects the amount of recovery for an ISI of 1s, obtained by extrapolating the fitted line. Higher visual areas show stronger RS and a slower recovery rate which is accurately predicted by the DN model. Data points indicate medians and error bars indicate 68% confidence interval across 1000 samples derived from the bootstrapped timecourses. Bootstrap test, * = $p < 0.05$ (two-tailed, Bonferroni-corrected). This figure can be reproduced by mkFigure5_6.py.

Fitting these responses with the DN model again shows accurate predictions: the model captures the overall gradual recovery from RS with longer time lags, closely mimicking the neural data (Fig 5A, lower panel and Fig 5B, DN model). The DN model also predicts stronger RS (Fig 6A, right), reflected in average level of suppression (Fig 6B, triangle markers) and faster recovery (Fig 6C, triangle markers), for higher than lower visual areas, although it underestimates the average suppression in VOTC and LOTC, possibly due to a slight over-prediction of the recovery for shorter ISIs. We also fitted neural responses with the A+S model and find that the A+S model poorly aligns with the neural data, predicting an overall higher degree of RS with area-dependent differences for short as opposed to long ISIs (S1(D) Fig).

Given prior reports of stimulus-specific differences in RS depending on a neural population's stimulus selectivity [3], we also quantified RS separately for preferred and non-preferred stimuli in all areas. In both neural responses and model predictions, the differences in RS

between areas are most pronounced for preferred stimuli (S4 Fig), and comparatively less strong for non-preferred stimuli (S5 Fig). This suggests that the repetition suppression effects in higher visual areas are partly stimulus-dependent.

## Differences in adaptation reflect slower normalization in higher visual areas

We showed that lower and higher visual areas show different adaptation patterns, as evident from transient-sustained dynamics and recovery from repetition suppression, which are both accurately captured by the DN model. To better understand the neural computations underlying these response profiles, we examined the temporal dynamics of two components of the DN model: the input drive (i.e. the numerator) and the normalization pool (i.e. the denominator).

To explain differences in transient-sustained dynamics, we considered the model prediction for the longest duration (533 ms, Fig 7A), because it has the most pronounced sustained response difference across areas. The DN model captures transient-sustained dynamics in neural responses because the input drive dominates the prediction early in the response, resulting in a transient, followed by the normalization pool, resulting in a response decay to sustained levels. The model suggests that lower visual areas exhibit relatively fast dynamics in both the

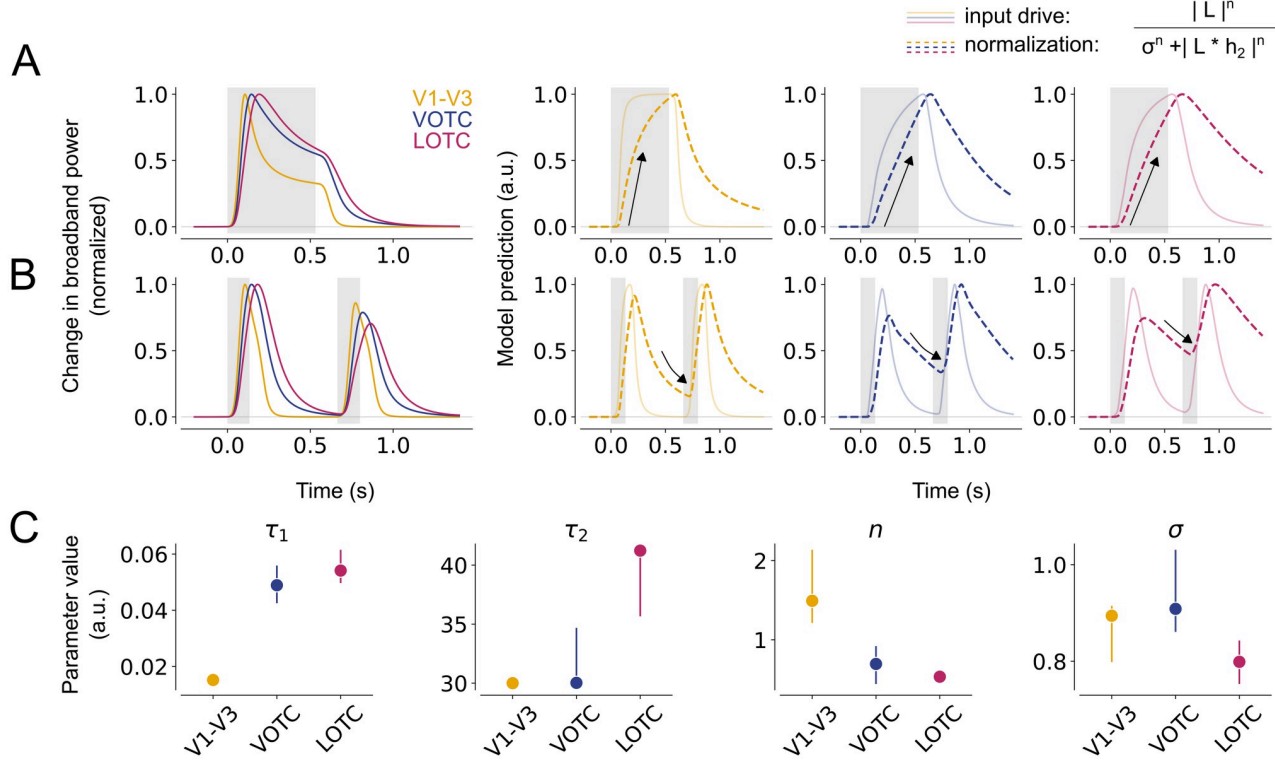

**Fig 7. Slower normalization in higher visual areas results in prolonged response shapes and stronger RS.** A. DN model prediction of the neural response for a single stimulus with a duration of 533 ms in V1-V3 (n = 17), VOTC (n = 15) and LOTC (n = 47). For each visual area, an additional panel is shown depicting the input drive (numerator, solid line) and the normalization pool (denominator, dashed line). The slower rise and prolonged response for VOTC and LOTC result from slower dynamics (arrow) of the normalization pool. B. Same as A for a repeated stimulus with an ISI of 533 ms. The stronger RS for higher visual areas results from lingering normalization at the start of the second stimulus, which is stronger for VOTC and LOTC compared to V1-V3. C. Fitted DN model parameters per visual area, from left to right: $h_1$ (time constant of the IRF), $h_2$ (time constant of the exponential decay), $n$ (exponent) and $\sigma$ (semi-saturation constant). Data points indicate medians and error bars indicate 68% confidence interval across 1000 samples derived from the bootstrapped timecourses. This figure can be reproduced by mkFigure7.py.

numerator and the denominator, resulting in a fast initial rise and a fast subsequent decay of the response. These dynamics occur at a lower pace in higher visual areas, where both the input drive and normalization pool rise more slowly. This results in broader response shapes, which are most pronounced for LOTC and to a lesser degree for VOTC.

To explain differences in recovery from RS, we again examined the longest temporal condition (ISI of 533 ms, Fig 7B), because differences in adaptation between lower and higher visual areas were most distinct at this ISI. The DN model captures suppression of repeated stimuli by adapting the dynamics of the normalization pool. After the offset of the first stimulus, the normalization pool decays and approaches the minimum possible value of the denominator, which is set by $\sigma^n$. If the normalization pool has not reached this minimum value at the start of the second stimulus, suppression occurs due to the lingering normalization from the first stimulus. Thus, the difference in RS between visual areas is a result of slower dynamics of the normalization pool in VOTC and LOTC, leading to more lingering normalization at the start of the second stimulus presentation and consequently stronger RS and slower recovery.

The differences in temporal adaptation across areas are also reflected in the fitted parameter values (Fig 7C). Both $\tau_1$ (time constant of the IRF) and $n$ (exponentiation) are higher in VOTC and LOTC, reflecting the slower dynamics of the input drive and normalization pool, which give rise to the area-dependent differences in transient-sustained dynamics and RS; $\tau_1$ controls the width of the transient, reflected by the time to peak, whereas $n$ controls the decay of the transient response. Thus, these parameters affect the full-width at half maximum and degree of recovery from RS for single and repeated stimuli, respectively. However, $\tau_2$ and $\sigma$ also affect the width and decay of the transient and fitted parameters (to some degree) trade off. Therefore, parameter differences across the visual hierarchy should be interpreted with caution. Nonetheless, our results suggest that adaptation differences between lower and higher visual areas could arise from underlying differences in temporal normalization dynamics.

### Stronger RS for preferred image categories in category-selective electrodes

The results indicated that transient-sustained dynamics are slower in higher than lower visual areas regardless of stimulus preference, whilst repetition suppression differences across areas are most pronounced for preferred stimuli (S4 Fig). To further investigate how adaptation is influenced by stimulus preference, we directly compared responses within a subset of electrodes in higher visual regions that exhibit strong category-selectivity.

We identified a subset of category-selective electrodes in LOTC and VOTC by calculating a sensitivity measure ($d'$) on the response per stimulus category averaged across all stimulus durations (see Materials and methods; for electrode positions and counts see Fig 1C and S4 Table, respectively). We then calculated average broadband responses separately for the preferred and non-preferred categories for each ISI and calculated recovery from RS similar as before. RS occurs for both preferred and non-preferred stimuli (Fig 8A, top panel), but more strongly for preferred stimuli (Fig 8B, Neural data). Model simulations show that the DN model also captures these differences, including the overall shape of the neural time courses (Fig 8A, bottom panel) and stronger RS for preferred stimuli (Fig 8B, DN model).

Quantifying the recovery from RS for the different stimulus types shows that the stronger RS for preferred image categories which is most pronounced for longer ISIs (Fig 8C, left), which is accurately captured by the model, although it slightly overestimating the degree of recovery for non-preferred stimuli for shorter ISIs (Fig 8C, right). Preferred stimuli show slower long-term recovery of RS (Fig 8D, circle markers) in both the neural data and the DN model (Fig 8D, triangle markers). These differences were robust in both data and model and became even more pronounced when increasing the threshold for category-selectivity

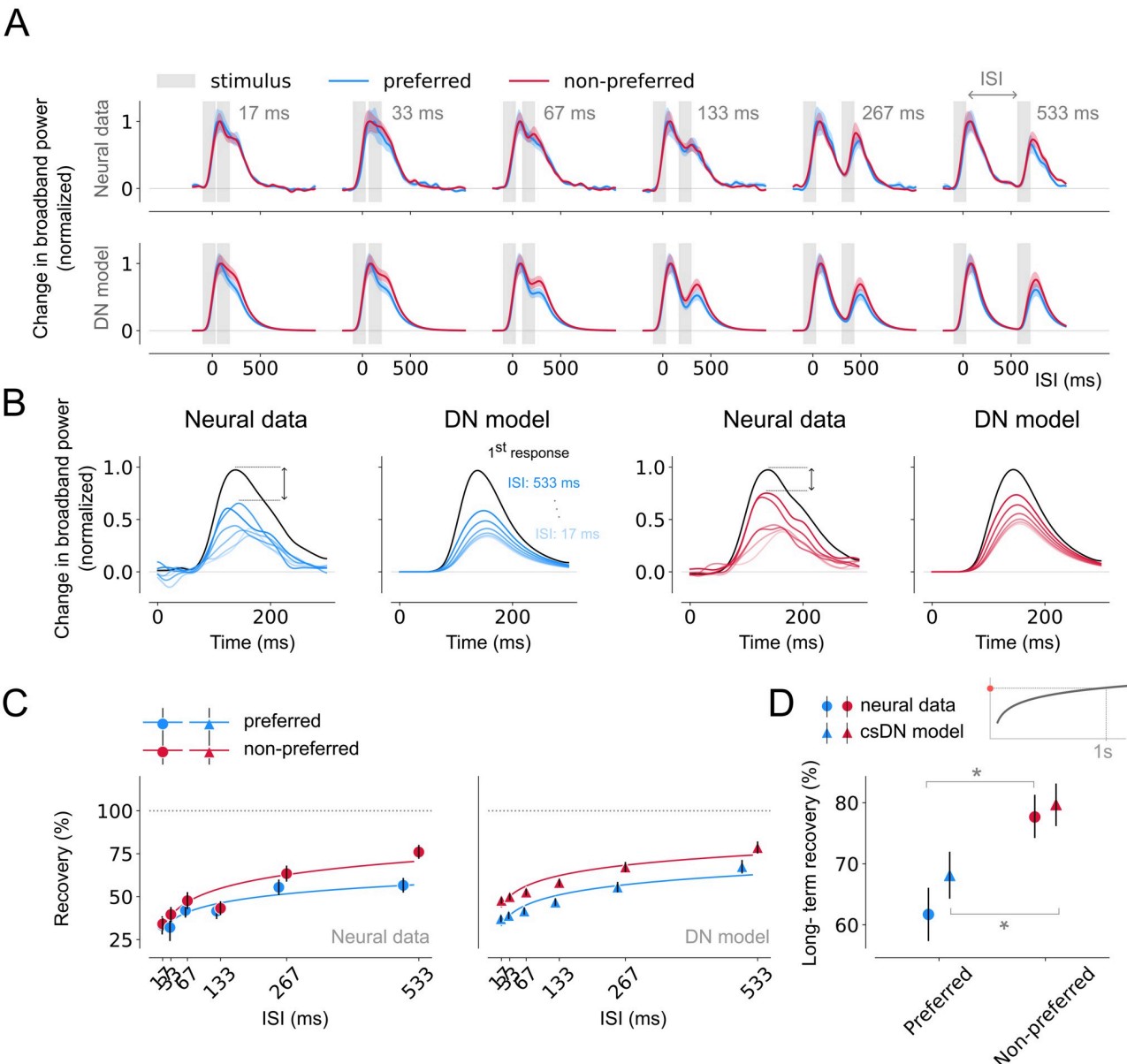

**Fig 8. Differences in recovery from adaptation across stimuli in category-selective areas.** A. Top, Average, normalized broadband responses of category-selective electrodes (threshold $d'$ = 0.5, n = 26) of trials during which preferred (blue) or non-preferred (red) stimuli were presented in repetition (gray). Responses are shown separately per ISI from shortest (17 ms, left) to longest (533 ms, right). Bottom, DN model predictions for the same data. Time courses differ for preferred and non-preferred stimuli which is captured by the DN model. For non-normalized responses see S6 Fig. B. Estimated, normalized response to the second stimulus for trials containing preferred and non-preferred stimuli. Per visual area, the left panel shows the neural data and the right panel shows the model prediction. The rate of recovery is higher for non-preferred compared to preferred stimuli. C: Recovery from adaptation computed as the ratio of the AUC between the first and second response derived from the neural data (left) or DN model predictions (right). The fitted curves express the degree of recovery as a function of the ISI (see Materials and methods, Summary metrics). Responses derived from trials containing preferred stimuli show a stronger degree of RS and the DN model is able to capture stimulus-specific recovery from adaptation. D: Long-term recovery from adaptation derived from the neural responses (circle marker) or DN model (triangle marker), reflecting the amount of recovery for an ISI of 1s. Responses for trials presenting preferred stimuli show stronger RS and a slower recovery rate. Data points indicate medians and error bars indicate 68% confidence interval across 1000 samples derived from the bootstrapped timecourses. Bootstrap test, * = $p < 0.05$ (two-tailed). This figure can be reproduced by mkFigure8.py.

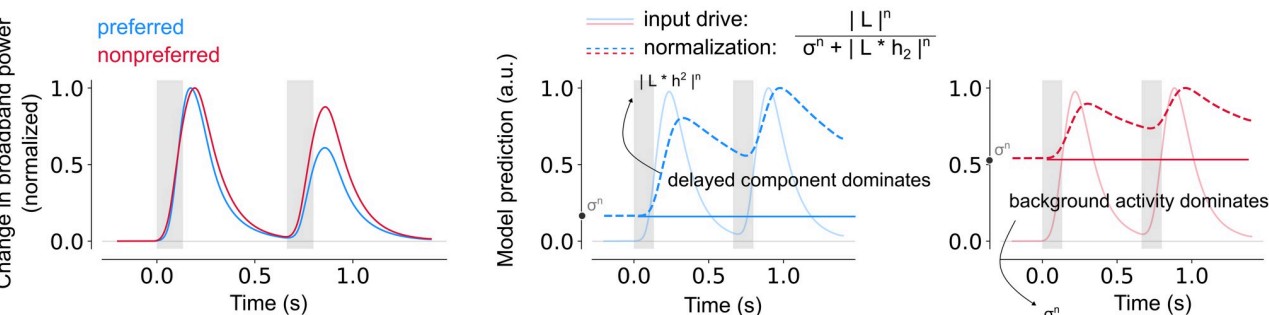

**Fig 9. Stronger RS for preferred stimuli in category-selective areas due to slower normalization dynamics and stronger input drive.** A. DN model predictions for repeated preferred (blue) or non-preferred (red) stimuli with an ISI of 533 ms. B. The input drive (solid lines) and normalization pool (dashed lines) plotted separately for a preferred (left) and non-preferred (right) stimulus. The stronger RS for preferred stimuli is a result of the higher lingering normalization at the start of the second stimulus, due to a larger input drive resulting in a delayed component amply surpassing the semi-saturation constant. Model time courses represented bootstrapping averages (n = 1000) across category-selective electrodes (threshold $d'$ = 0.5, n = 26). This figure can be reproduced by mkFigure9.py.

selection (see S8 and S9 Figs for a threshold of $d'$ of 0.75 and 1, respectively, resulting in fewer selected electrodes).

## Lingering normalization and stronger input drive result in stronger adaptation and slower recovery rate for preferred stimuli

Our results suggest that preferred stimuli elicit stronger RS than non-preferred stimuli in category-selective electrodes in VOTC and LOTC. The DN model explains this from the balance between the two components that make up the model denominator (Fig 9). As before, RS for both stimulus types results from lingering normalization at the start of the second stimulus. For preferred stimuli, the input drive is strong, and therefore the lingering normalization amply surpasses the value of the semi-saturation constant, $\sigma^n$. Because dynamics are slow, the lingering normalization is (relatively) high at the start of the second response, resulting in strong RS. For non-preferred stimuli, the lingering normalization is much smaller in comparison to $\sigma^n$, due to the weaker input drive. While there is still lingering activity at the start of the second stimulus, $\sigma^n$ comprises a much larger part of the denominator, marginalizing the effect of the lingering normalization. Since $\sigma^n$ is the same for the first and second stimulus, less RS is observed.

In short, the differences in adaptation between preferred and non-preferred stimuli in category-selective electrodes can be explained by the balance between the normalization pool components, which depends on the initial input drive, in combination with the slower dynamics in higher visual areas.

## Discussion

Our aim was to examine how short-term neural adaptation differs across human visual cortex and to pinpoint the underlying neural computations using a model of delayed divisive normalization. We demonstrate that, compared to V1-V3, higher visual areas have more prolonged responses for single stimuli and stronger repetition suppression for repeated stimuli. The DN model accurately predicts the neural response time courses and their adaptation profiles in both lower and higher visual areas by means of a category-dependent scaling on the input stimulus time course. The model fits show that differences in temporal adaptation across areas can be explained by slower dynamics of both the input drive and normalization pool for higher

visual regions. We additionally find that neural responses in category-selective electrodes exhibit stronger RS for preferred than non-preferred stimuli, which the DN model explains from the balance of the normalization pool components in combination with slower dynamics in these regions.

We believe this study offers several novel insights. First, we demonstrate clear differences in temporal dynamics between lower and higher visual areas when using naturalistic stimuli that drive both low and high-level regions well. Earlier studies which examined temporal dynamics across lower and higher areas used stimuli that mostly drive lower visual regions. Our results show that stimulus effectiveness affects short-term adaptation in various ways and should therefore be carefully considered when measuring and modeling adaptation across the visual hierarchy. Second, while previous work using single-cell recordings to study stimulus-specific effects on temporal dynamics of neural responses across the visual cortex, this study is to our knowledge the first to demonstrate and model the neural computations possibly underlying such effects in human data with both high spatial- and temporal resolution.

## Slower time-scales of neural processing in higher visual areas

We observed prolonged responses with slower transient-sustained dynamics in higher visual areas. This is consistent with the idea that time scales of temporal processing become longer when ascending the visual hierarchy, as suggested based on brain responses to both single [7, 14] and repeated stimulus presentations [7, 13, 26, 29], as well as the pattern of responses to intact and scrambled natural movies [46, 47]. Increasing temporal windows across the cortical hierarchy may have several computational benefits. First, [48] proposed that such a hierarchy is useful for prediction over multiple timescales. Second, temporal windows may be tuned to the temporal regularities of the input features, as demonstrated in both theoretical [49] and empirical work [46, 47]. Different types of image feature are likely to exhibit different temporal regularities in natural viewing conditions: low-level features (e.g., orientation, edges, and contrast) change each time an observer moves their eyes, thereby benefiting from shorter processing windows, while high-level features (e.g., holistic representations of faces and objects) are likely to be stable over longer viewing durations, and areas tuned to that information may therefore be tuned to longer timescales. In addition to computational benefits, the ability to integrate and hold information across a variety of time scales is also critical for cognition and flexible behaviour [50].

In addition to a hierarchy within unimodal areas (e.g. visual or auditory cortex), there may also be a hierarchy of time scales in multimodal processing, with shorter time windows in unimodal regions and longer time windows in association cortex (e.g. lateral prefrontal cortex or the default model network), which has been observed across several acquisition modalities, species and task states (e.g. [51]). It is believed that this hierarchy of timescales plays a key role in both integrating and segregating sensory information across time. Regions with shorter timescales may favour temporal segregation, reflected by shorter neural responses, whereas higher areas are involved in temporal integration, reflected by longer neural responses. This balance of temporal integration and segregation may enable the segmentation of continuous inputs (for a review see [52]), benefiting perception and cognition. Whether similar distinctions can be made between lower and higher regions within unimodal areas in visual cortex, and how this contributes to perception, warrants future investigation.

## Slower recovery from RS in higher visual areas

We found differences in the overall degree of repetition suppression and recovery rate from RS between lower and higher visual areas. These results differ from a prior study [14], which

found that the degree of RS and the recovery rate from RS did not differ between early visual and lateral-occipital retinotopic regions, ranging from V1 to IPS. Here we find stronger RS as well as higher recovery rates in VOTC and LOTC compared to V1-V3. We attribute the difference between studies to the difference in stimuli, simple contrast patterns in [14] vs naturalistic stimuli in this study. Simple contrast patterns strongly drive responses in lower visual areas (V1-hV4, [7, 53]), but not higher areas that are selective for complex, naturalistic stimuli [54–56]. The reduced responses in higher areas to simple contrast patterns could have made it more difficult to accurately measure RS. In addition to eliciting weaker responses, sub-optimal stimuli may have also led to less RS in higher areas, making the adaptation patterns more similar to early areas. This explanation is supported by our current observations of similar RS between areas for non-preferred stimuli (S5 Fig), as well as less RS for non-preferred stimuli within category-selective electrodes (Fig 8). Furthermore, stimulus type influences not only the magnitude of neural responses but also their temporal stability [57] as well as their oscillatory components [58], which could also affect RS patterns.

An fMRI study on short-term adaptation by [26] found stronger RS for higher visual regions, consistent with our findings, but did not observe differences in recovery rate between visual areas despite using complex stimuli, differing from our findings. One reason for the discrepancy with our findings could be the way (recovery from) adaptation was computed. As the sluggish nature of the BOLD signal makes it difficult to estimate independent fMRI responses to stimuli presented close in time, [26] used stimulus pairs consisting of either repeated, identical stimuli, or two distinct stimuli, and quantified RS as the difference in the maximal response to identical versus non-identical stimulus pairs. In contrast, we measured iEEG responses only to repeated representations of the same image, and measured recovery from RS as the difference in response AUC between the first and second stimulus representation.

## Differences in temporal dynamics between ventral and lateral occipital cortex

We separated our electrodes into two higher-level groups covering ventral and lateral occipitotemporal cortex, respectively. Previous work has shown differences in the temporal dynamics between these regions using an encoding framework where neural responses were modelled in separate sustained and transient channels [44]. VOTC responded to both transient and sustained visual inputs, while LOTC predominantly responded to visual transients. The authors suggested that VOTC regions are mainly involved in processing of static inputs while LOTC regions process dynamic inputs. In contrast, our data show a more sustained response in LOTC compared to VOTC (Fig 4C). These sustained responses could indicate that LOTC accumulates information over relatively longer time periods, in line with work suggesting that LOTC regions may also be involved in more stable information processing [47].

While [44] showed that VOTC and LOTC both exhibit transient responses, they also observed differences in the dynamics of transient processing across the two visual streams. In LOTC, the onset and offset of the visual stimulus elicited equal increase in neural responses, suggesting that these areas process information regarding moment-to-moment changes in the visual input. In VOTC, the transient responses for the onset and offset of the stimulus were surprisingly asymmetric and were mostly dominated by stimulus offset. The authors hypothesized that this reflected memory traces maintained by these regions after the stimulus is no longer visible. In our data however, we did not observe strong stimulus offset responses. A similar lack of offset responses was observed in earlier ECoG studies [13, 14]. [13] furthermore noted that offset responses were more pronounced for electrodes with peripherally tuned spatial receptive fields (beyond 5 degrees eccentricity). The stimuli used in the current study

extended to 8.5 eccentricity, therefore the lack of offset responses may be related to the spatial coverage of the stimulus. However, other explanations are possible, such as differences in data type (fMRI vs. ECoG), brain areas sampled, or experimental design. In conclusion, further research is needed to elucidate the differences in temporal dynamics between higher-level regions and how they relate to the timescales of the visual input.

### Stimulus-specific differences in temporal dynamics in category-selective areas

We found stronger RS for preferred than non-preferred stimuli in category-selective electrodes, consistent with findings from fMRI [29], single-cell recordings [3, 59] and ECoG [30]. The DN model shows that the stimulus-specific adaptation differences could result from the balance in normalization pool components in combination with slower normalization dynamics in these areas. The strong input drive for preferred stimuli causes more lingering normalization so that when the second stimulus arrives, there is a reduced response. To model effects of stimulus preference on neural response dynamics, we augmented the DN model by incorporating category-dependent scaling on the stimulus timecourse. Model fits showed that adding a category-based scaling factor results in better predictions in all visual regions, including V1-V3, which is not typically considered to exhibit category-selectivity. We attribute the scaling benefit in these early visual regions to co-variation of low-level feature differences with the categories in the dataset. Specifically, one of the six categories consisted of scrambled stimuli which had many edge elements, and one of scene stimuli which had a slightly larger retinotopic extent than the other classes. These classes likely are more optimal stimuli for lower visual areas.

While our data revealed stimulus-specific effects on RS for repeating stimuli, we observe weak to no effect of stimulus preference on transient-sustained dynamics during single stimulus trials. This is in line with previous work on non-human primates using single-cell recordings, which predominantly report differences in temporal dynamics in the context of RS [3, 22, 60–62]. While some studies also present stimuli in isolation, they do not further examine adaptation-related differences based on stimulus preference. For example [3] showed, similar to our experimental paradigm, stimulus sequences with either identical or varying images that elicited weaker or stronger responses depending on stimulus preference. While the authors do make comparisons between repeated stimuli and single stimuli, no analysis is conducted regarding the dynamics of preferred and non-preferred stimuli in isolation. While the lack of reports regarding stimulus-dependent effects on transient-sustained dynamics does not evince their nonexistence, further research should elucidate the presence of stimulus-specific effects on the temporal dynamics during briefly presented stimuli with varying durations.

### Limitations and future work

First, since electrode positioning was determined based on clinical constraints, the number of electrodes localized to individual retinotopic maps was limited. Therefore, our comparisons focused on coarse groupings of the visual areas: early (V1-V3) versus ventral (VOTC) versus lateral (LOTC) maps. For fine-grained comparison between visual areas across the cortical hierarchy (say V1 vs V2), different methods are needed. Related to this, there was an imbalance in the area-wise distribution of electrodes across subjects, with only two subjects contributing electrodes covering early visual maps (S3 Table). Interpretation of the results in the context of the visual hierarchy should therefore be made with caution. Second, the current model form does not explicitly represent the computations in each stage of processing, and so the model is agnostic to the origin of the divisive signals. Third, the behavioral task

participants performed was orthogonal to the temporal stimulus manipulations. This design was purposeful to reduce variability in top-down signals from trial to trial and between participants. Nonetheless, neural adaptation is important for behavior such as priming [60, 63], and the link between them cannot be directly studied without a task that is relevant to the stimulus.

Several approaches could be undertaken to tackle some of these limitations, including collecting and fitting behavioral measurements of adaptation with the DN model, or measuring transient-sustained dynamics and RS in neural data from animals to allow a more systematic comparison across the visual hierarchy. Another approach is to study adaptation in Artificial Neural Networks (ANNs). ANNs have recently come forward as a powerful new tool to model sensory processing [64–66]. These models are image-computable, are trained to process naturalistic stimuli, consist of units whose activations are inspired by biological neuronal signals, and output predictions that can be compared with human behavior. Moreover, these models process inputs in a sequential fashion, where activations from earlier layers are fed to later layers which is comparable to the input-out transformations mimicking the neural processing from lower- to higher-level areas. Future studies could examine the link between adaptation phenomena and behavior by implementing biologically plausible adaptation in ANNs, including divisive normalization. Such paradigms could aid in better understanding how different adaptation mechanisms may benefit perception.

Lastly, we would like to note that the DN model as presented in this study, is not a circuit-level model and the predicted neural responses can be the result of a variety of biophysical and cellular mechanisms. Future studies should perform a more in-depth examination, using other types of data such as single-cell recordings and alternative models (e.g. [67]), to identify the neural circuitry that could give rise to observed normalization dynamics across visual areas and stimuli.

## Supporting information

**S1 Fig. The two-temporal channel model with adaptation and sigmoidal nonlinearities (A +S, [44]) fails to capture transient-sustained dynamics and repetition suppression (RS) observed in neural responses.** A: Cross-validated explained variance (coefficient of determination) across all stimulus conditions for the DN model compared with a two-channel model from [24] (linear + quadratic, L+Q) and [44] (adaptation + sigmoid, A+S) plotted per visual area (V1-V3, VOTC and LOTC). Category-selective scaling is either omitted or included during model fitting. The DN model predicts neural responses to a higher degree compared to both implementations of the two-channel models. Category-dependent scaling further improves model fits. B: Top, Average, normalized broadband iEEG responses (80–200 Hz) for electrodes assigned to V1-V3 (n = 17), VOTC (n = 15) and LOTC (n = 47) to single stimuli (gray). Responses are shown separately for a stimulus duration of 267 and 533 ms. The TTC model predicts an offset response which is not present in the neural data (black arrow). C: Full-width at half maximum, computed for each stimulus duration. The TTC model predicts narrower response widths for VOTC compared to what is observed in the neural data. D: left, Recovery from adaptation computed as the ratio of the Area Under the Curve (AUC) between the first and second response derived from the neural data. The fitted curves express the degree of recovery as a function of the ISI (see Materials and methods, Summary metrics). Right, Same as left for the TTC model. The TTC model poorly aligns with the neural data and predicts an overall higher degree of recovery from RS with area-dependent differences for short as opposed to long ISIs. Data points indicate medians and error bars indicate 68% confidence interval across 1000 samples derived from the bootstrapped timecourses. Panel A can be

reproduced by mkSuppFigure1.py. Panel BC and D can be reproduced by mkSuppFigure4.py and mkFigure5_6.py, respectively.
(PDF)

**S2 Fig. Slower rise and prolonged responses in higher visual areas for single stimuli for preferred and non-preferred stimuli.** (*previous page*). A: Top, Average, normalized broadband iEEG responses (80–200 Hz) for electrodes assigned to V1-V3 (n = 17), VOTC (n = 15) and LOTC (n = 47) to single, preferred stimuli (gray). Responses are shown separately per duration from shortest (17ms, left) to longest (533 ms, right). Bottom, DN model predictions for the same conditions. The shapes of the neural time courses differ between visual areas and are accurately captured by the DN model. Time courses were smoothed with a Gaussian kernel with standard deviation of $\sigma = 10$; the shaded regions indicate 68% confidence interval across 1000 bootstrapped timecourses (see Materials and methods, Bootstrapping procedure and statistical testing). B-C: Summary metrics plotted per visual area derived from the neural responses (circle marker) or model time courses (triangle marker). Time-to-peak (B) computed to the longest duration (533 ms). Full-width at half maximum (C), computed for each stimulus duration. For higher visual areas, results suggest that neural responses show stronger reduction at stimulus offset for preferred compared to non-preferred stimuli (black arrow in panel A and D), which is captured by the DN model. Data points indicate medians and error bars indicate 68% confidence interval across 1000 samples derived from the bootstrapped timecourses. Bootstrap test, * = $p < 0.05$ (two-tailed, Bonferroni-corrected). D-F: Same as A-C for trials showing non-preferred stimuli. This figure can be reproduced by mkFigure4.py.
(PDF)

**S3 Fig. Extract degree of recovery from repetition suppression.** A. Top, the broadband time course of an example electrode averaged over all repetition trials plotted separately for each ISI. Bottom, an estimate of the response to the first and second stimulus. A robust response to the first stimulus is obtained by averaging the response time courses for the 134 ms duration stimulus and each of the repetition stimuli from trial onset up to the onset of the second stimulus. The estimate of the response to the second stimulus is computed by subtracting the average time course of the first stimulus from the ISI varying stimulus responses. B: The recovery from neural adaptation is computed and defined as the Area Under the Curve (AUC) of the second pulse proportional to AUC of the first pulse, as a function of the ISI. This figure can be reproduced by mkSuppFigure3.py.
(PDF)

**S4 Fig. Differences in recovery from repetition suppression across visual areas for preferred stimuli.** A. Estimated, normalized response to the second stimulus for V1-V3, VOTC and LOTC. For each visual area, the left panel shows the neural data and the right panel shows the model prediction. Recovery from adaptation gradually increases as the ISI becomes longer, and the rate of recovery is higher for V1-V3 compared to VOTC and LOTC in both the neural data and the DN model. B: left, Recovery from adaptation for V1-V3 (n = 17), VOTC (n = 15) and LOTC (n = 47), computed as the ratio of the Area Under the Curve (AUC) between the first and second response. The fitted curves express the degree of recovery as a function of the ISI (see Materials and methods, Summary metrics). Higher visual areas show stronger RS and slower recovery from adaptation. Right, model predictions for the same data. The model is able to capture area-specific recovery from adaptation. B-C: Summary metrics plotted per visual area derived from the neural responses (circle marker) or model time courses (triangle marker). Average recovery (B) from adaptation for each area, computed by averaging the AUC ratios between the first and second stimulus over all ISIs. The long-term recovery (C)

reflects the amount of recovery for an ISI of 1s, obtained by extrapolating the fitted line. Higher visual areas show stronger RS and a slower recovery rate which is accurately predicted by the DN model. Data points indicate medians and error bars indicate 68% confidence interval across 1000 samples derived from the bootstrapped timecourses. Bootstrap test, * = $p < 0.05$ (two-tailed, Bonferroni-corrected). This figure can be reproduced by mkFigure5_6. py.
(PDF)

**S5 Fig. Differences in recovery from repetition suppression across visual areas for non-preferred stimuli.** A. Estimated, normalized response to the second stimulus for V1-V3, VOTC and LOTC. For each visual area, the left panel shows the neural data and the right panel shows the model prediction. Time courses were obtained using a bootstrapping procedure ($n = 1000$, see Materials and methods, Bootstrapping procedure and statistical testing). Recovery from adaptation gradually increases as the ISI becomes longer, and the rate of recovery is higher for V1-V3 compared to VOTC and LOTC in both the neural data and the DN model. B: left, Recovery from adaptation for V1-V3 (n = 17), VOTC (n = 15) and LOTC (n = 47), computed as the ratio of the Area Under the Curve (AUC) between the first and second response. The fitted curves express the degree of recovery as a function of the ISI (see Materials and methods, Summary metrics). Higher visual areas show stronger RS and slower recovery from adaptation. Area-related differences are less pronounces compared to preferred stimulus trials (S5 Fig). Right, model predictions for the same data. The model is able to capture area-specific recovery from adaptation. B-C: Summary metrics plotted per visual area derived from the neural responses (circle marker) or model time courses (triangle marker). Average recovery (B) from adaptation for each area, computed by averaging the AUC ratios between the first and second stimulus over all ISIs. The long-term recovery (C) reflects the amount of recovery for an ISI of 1s, obtained by extrapolating the fitted line. Higher visual areas show stronger RS and a slower recovery rate which is accurately predicted by the DN model. Data points indicate medians and error bars indicate 68% confidence interval across 1000 samples derived from the bootstrapped timecourses. Bootstrap test, * = $p < 0.05$ (two-tailed, Bonferroni-corrected). This figure can be reproduced by mkFigure5_6.py.
(PDF)

**S6 Fig. Differences in recovery from adaptation across stimuli in category-selective areas.** A. Top, Average, broadband responses of category-selective electrodes (threshold $d' = 0.75$, n = 12) of trials during which preferred (blue) or non-preferred (red) stimuli were presented in repetition (gray). Time courses were obtained using a bootstrapping procedure ($n = 1000$, see Materials and methods, Bootstrapping procedure and statistical testing). Responses are shown separately per ISI from shortest (17 ms, left) to longest (533 ms, right). Bottom, DN model predictions for the same data. Time courses differ for preferred and non-preferred stimuli which is captured by the DN model. This figure can be reproduced by mkFigure8.py.
(PDF)

**S7 Fig. Differences in recovery from adaptation across stimuli in category-selective areas.** A. Top, Average, normalized broadband responses of category-selective electrodes (threshold $d' = 0.75$, n = 12) of trials during which preferred (blue) or non-preferred (red) stimuli were presented in repetition (gray). Time courses were obtained using a bootstrapping procedure ($n = 1000$, see Materials and methods, Bootstrapping procedure and statistical testing). Responses are shown separately per ISI from shortest (17 ms, left) to longest (533 ms, right). Bottom, DN model predictions for the same data. Time courses differ for preferred and non-preferred stimuli which is captured by the DN model. B. Estimated, normalized response to

the second stimulus for trials containing preferred and non-preferred stimuli. Per visual area, the left panel shows the neural data and the right panel shows the model prediction. The rate of recovery is higher for non-preferred compared to preferred stimuli. C: Recovery from adaptation computed as the ratio of the AUC between the first and second response derived from the neural data (left) or DN model predictions (right). The fitted curves express the degree of recovery as a function of the ISI (see Materials and methods, Summary metrics). Responses derived from trials containing preferred stimuli show a stronger degree of RS and the DN model is able to capture stimulus-specific recovery from adaptation. D: Long-term recovery from adaptation derived from the neural responses (circle marker) or DN model (triangle marker), reflecting the amount of recovery for an ISI of 1s. Responses for trials presenting preferred stimuli show stronger RS and a slower recovery rate. Data points indicate medians and error bars indicate 68% confidence interval across 1000 samples derived from the bootstrapped timecourses. Bootstrap test, $^*$ = $p < 0.05$ (two-tailed). This figure can be reproduced by mkFigure8.py.
(PDF)

**S8 Fig. Differences in recovery from adaptation across stimuli in category-selective areas.** A. Top, Average, normalized broadband responses of category-selective electrodes (threshold $d' = 1$, n = 6) of trials during which preferred (blue) or non-preferred (red) stimuli were presented in repetition (gray). Responses are shown separately per ISI from shortest (17 ms, left) to longest (533 ms, right). Bottom, DN model predictions for the same data. Time courses differ for preferred and non-preferred stimuli which is captured by the DN model. B. Estimated, normalized response to the second stimulus for trials containing preferred and non-preferred stimuli. Per visual area, the left panel shows the neural data and the right panel shows the model prediction. The rate of recovery is higher for non-preferred compared to preferred stimuli. C: Recovery from adaptation computed as the ratio of the AUC between the first and second response derived from the neural data (left) or DN model predictions (right). The fitted curves express the degree of recovery as a function of the ISI (see Materials and methods, Summary metrics). Responses derived from trials containing preferred stimuli show a stronger degree of RS and the DN model is able to capture stimulus-specific recovery from adaptation. D: Long-term recovery from adaptation derived from the neural responses (circle marker) or DN model (triangle marker), reflecting the amount of recovery for an ISI of 1s. Responses for trials presenting preferred stimuli show stronger RS and a slower recovery rate. Data points indicate medians and error bars indicate 68% confidence interval across 1000 samples derived from the bootstrapped timecourses. Bootstrap test, $^*$ = $p < 0.05$ (two-tailed). This figure can be reproduced by mkFigure8.py.
(PDF)

**S9 Fig. Electrode positions for individual participants, overlaid on a retinotopic atlas.** (*previous page*). Electrode positions for subject p11 (A), subject p12 (B), subject p13 (C) and subject p14 (D) overlaid on a pial surface reconstruction with colour-coded predicted visual locations. A surface node in the pial mesh was assigned a colour if it had a non-zero probability of being in a visual region according to a max probability map from [42]. If the electrode had a nonzero probability of being in multiple regions, the region with the highest probability was assigned. For visualization purposes, some retinotopic maps have been merged (e.g., dorsal and ventral parts of V1). The brain surfaces were created using Freesurfer [39] and scripts can be found at https://github.com/WinawerLab/ECoG_utils. L = lateral, M = medial, D = dorsal, V = ventral, A = anterior, P = posterior.
(PDF)

**S10 Fig. Electrode positions.** A. Electrodes with robust visual responses were assigned to early (V1-V3, n = 17), VOTC (n = 15) or LOTC (n = 47) retinotopic areas. Electrodes that were not included in the dataset are shown in black. Electrodes were considered category-selective if the average response for a given image category was higher compared to the other image categories ($d' > 0.75$, see Eq 1, Materials and methods, n = 12). B. Same as A for a threshold of $d'$ for category-selectivity of 1.0 (n = 6). The brain surfaces were created using MNE-Python and can be reproduced by mkFigure2.py. L = lateral, M = medial, D = dorsal, V = ventral, A = anterior, P = posterior.
(PDF)

**S11 Fig. DN model without category-dependent input strength has poor fits for non-preferred image categories.** DN model prediction (red) of the neural response (black) of an example electrode, given a stimulus timecourse (light blue), with (top panel) and without (bottom panel) category-specific scaling. Model prediction for the least preferred image category (faces) results in an overestimation of the neural response, resulting in a strongly negative coefficient of determination (blue arrow). This figure can be reproduced by mkSuppFigure11.py.
(PDF)

**S1 Table. Overview of patient data included in this dataset.** Columns refer to the following: Subject, subject code in dataset. Age, age of patient at time of recording in years. Sex, gender of the participant. Implantation, type of electrodes implanted. Grid, standard clinical grid; HDgrid, high-density grid; strip, standard clinical strip; depth, depth electrodes. Runs, number of runs where a run is defined as a period of sequential stimulus presentations with no breaks in between. Trials, number of trials collected where half consisted of duration and half consisted of repetition trials (e.g. for sub-p11, there were 432 duration and 432 repetition trials). Repetitions, number of times a stimulus set was repeated (separate stimuli were used for even and uneven runs).
(PDF)

**S2 Table. Overview of visual areas included in this dataset.** Columns refer to the following: Visual areas, visual areas to which electrodes are assigned, V1-V3, early visual cortex; VOTC, ventral-occipital cortex; LOTC: lateral-occipital cortex. Matching probabilistic areas, visual areas according to the maximum probability atlas by [42]. Matching retinotopic areas, visual areas according to an anatomically defined atlas by [40] and [41].
(PDF)

**S3 Table. Overview of electrodes included and visual areas covered in this dataset.** Columns refer to the following: Subject, subject code in dataset. Electrodes, total number of electrodes. Visual areas, visual areas to which electrodes are assigned, V1-V3, early visual cortex; VOTC, ventral-occipital cortex; LOTC: lateral-occipital cortex. The number of electrodes per area is reported within the parentheses. Matching areas, visual areas included according to the maximum probability atlas by [42] (left column) or a retinotopic atlas developed by [40] and [41] using a Bayesian mapping approach (right column). Visually responsive electrodes, the number of electrodes assigned according by one of the atlases (left column) or assigned manually (right column) to V1-V3, VOTC or LOTC.
(PDF)

**S4 Table. Overview of category-selective electrodes.** Columns refer to the following: $d'$ threshold, threshold for an electrode to be considered category-selective (see Eq 1, Materials and methods). Image categories, number of electrodes selected per image category. The number of electrodes per area is reported within the parentheses. Total, total number of category-

selective electrodes for the specified $d'$ threshold.
(PDF)

## Author Contributions

**Data curation:** Amber Marijn Brands, Iris Isabelle Anna Groen.

**Formal analysis:** Amber Marijn Brands, Iris Isabelle Anna Groen.

**Funding acquisition:** Sasha Devore, Orrin Devinsky, Adeen Flinker, Jonathan Winawer, Iris Isabelle Anna Groen.

**Methodology:** Amber Marijn Brands, Jonathan Winawer, Iris Isabelle Anna Groen.

**Project administration:** Sasha Devore, Orrin Devinsky, Werner Doyle, Adeen Flinker, Jonathan Winawer, Iris Isabelle Anna Groen.

**Resources:** Sasha Devore, Orrin Devinsky, Werner Doyle, Adeen Flinker, Daniel Friedman, Patricia Dugan.

**Software:** Amber Marijn Brands, Iris Isabelle Anna Groen.

**Supervision:** Iris Isabelle Anna Groen.

**Visualization:** Amber Marijn Brands, Jonathan Winawer.

**Writing – original draft:** Amber Marijn Brands, Iris Isabelle Anna Groen.

**Writing – review & editing:** Orrin Devinsky, Adeen Flinker, Jonathan Winawer.

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
