## [Decision Letter · Decision Letter 0]

30 Nov 2023

Dear Miss Brands,

Thank you very much for submitting your manuscript "Temporal dynamics of neural adaptation across human visual cortex" for consideration at PLOS Computational Biology.

As with all papers reviewed by the journal, your manuscript was reviewed by members of the editorial board and by several independent reviewers. In light of the reviews (below this email), we would like to invite the resubmission of a significantly-revised version that takes into account the reviewers' comments.

The reviewers express a number of important issues regarding the paper, please make sure to address them all in case of a revised version. In particular, we agree with the concerns they raise about the lack of comparison with other models (other forms of normalization, and other models proper) and the degree of novelty of the reported results.

We cannot make any decision about publication until we have seen the revised manuscript and your response to the reviewers' comments. Your revised manuscript is also likely to be sent to reviewers for further evaluation.

Sincerely,

Hugues Berry

Academic Editor

PLOS Computational Biology

Daniele Marinazzo

Section Editor

PLOS Computational Biology

Reviewer's Responses to Questions

**Comments to the Authors:**

Reviewer #1: The visual world is highly dynamic. In the dynamic realm of visual perception, the temporal characteristics of cortical neurons play an indispensable role. Researchers have developed computational models to more accurately describe how visual cortex neurons encode dynamic information. There are two popular computational models describing the encoding of temporally dynamic perceptual information by populations of visual cortex neurons. (1) The two-temporal-channel (TTC) model (Horiguchi, Nakadomari, Misaki, & Wandell, 2009; Stigliani, Jeska, & Grill-Spector, 2017, 2019) explains the cortical dynamics by the response properties of transient and sustained neuronal populations. (2) The delayed normalization (DN) model (Groen et al., 2022; Zhou, Benson, Kay, & Winawer, 2019), which is built by authors in the current study, tries to explain the temporal dynamics of neural signals in human visual cortex using a unified computational function. The core idea of the DN model is that the stimulus that drives neuronal population activity is divisively normalized by delayed neuronal population activity (delayed via a low-pass filter). The DN model predicts many temporal phenomena in human visual cortex, such as saturation as stimulus contrast and duration increase, suppression for repeated stimuli at short intervals, and slower onsets at low contrast (Groen et al., 2022; Zhou et al., 2019).

The current study builds upon the authors' extensive and commendable efforts in developing the DN model. In a groundbreaking move, Brands and colleagues have integrated a category-selectivity feature module into the original DN model, markedly enhancing its explanatory power. This addition not only elevates the biological interpretability of the DN model but also establishes this work as a pivotal achievement in the realm of visual dynamic modeling. However, my appreciation for this research is tempered by certain limitations that I have observed:

Major:

1. In one of the authors' previous works (Groen et al., 2022), an insightful comparison was provided between the DN model and the TTC model. Building on this, I am interested in understanding how the results might differ if the enhanced DN model, now incorporating category selectivity, were compared to a TTC model that similarly employs a scaling factor for modulating input strength. Such a comparison could offer a more comprehensive understanding of the relative strengths and applications of these two models under modified conditions, and would strongly strengthen the findings in the current study.

2. This study presents an interesting finding that repetition suppression effects differ across preferred and non-preferred image categories. The authors explain these variations in the "Stimulus-specific differences in temporal dynamics in category-selective areas" section of the Discussion, attributing them to a combination of normalization pool component balances and slower normalization dynamics in category-selective areas. However, contrasting observations are noted in the "Higher visual areas exhibit slower and prolonged responses to single stimuli" section and Supplementary Figure 1, showing no such variations in transient-sustained dynamics. The authors propose that this could be due to inherently slower temporal integration in the transient-sustained dynamics of higher visual regions. Given these observations, it would be beneficial for the authors to provide a more detailed explanation of how these findings align within a unified theoretical framework. Particularly, it would be insightful to explore how category selectivity influences the two forms of temporal adaptation differently. Expanding on this point in the discussion, possibly including a comparative analysis with findings from non-human primate studies, could significantly deepen and broaden the scope of the current study's conclusions.

3. The neuronal response to stimulus offset is a significant aspect of neurophysiological studies, as evidenced by relevant research such as Stigliani et al. (2019), which highlighted the importance of offset response. In this context, I noticed that the current study does not include an analysis of the offset response, either in the neural or model responses. Could the author provide an explanation for this omission? Understanding the rationale behind excluding the offset response would not only clarify the scope of the study but also help in comprehensively evaluating its findings in the context of existing literature.

4. It's well-documented that the V1-V3 areas respond to stimuli differently compared to higher visual areas. Specifically, while higher visual areas show significant category selectivity, the V1-V3 areas are more attuned to low-level feature selectivity. Given this distinction, I am keen for the authors to clarify how integrating a category-selectivity-based scaling factor into the model's input strength specifically benefits its performance in the V1-V3 regions. Moreover, it would be insightful if the study examined the potential enhancement of the model's predictive performance through the inclusion of low-level feature selectivity parameters, such as orientation or color. An exploration in this direction could provide a more comprehensive understanding of the model’s effectiveness across various visual processing areas and its capability to encompass a wider range of neuronal responses.

5. The study currently does not report the specific number of category-selective electrodes present in the VOTC and the LOTC. For a more comprehensive understanding of the analysis, it is crucial to have detailed information regarding the proportion of category-selective electrodes in VOTC and LOTC.

6. In the Materials and Methods section, the authors outline "two consecutive data selection steps". These steps include an initial normalization of the signal to a percentage signal change, which is then followed by determining responsiveness using a z-score threshold, effectively serving as a second normalization process. Could the authors elaborate on the reasoning behind the use of two different normalization techniques in this sequential manner?

7. The manuscript lacks detailed explanations on how the scaling factor is computed. Furthermore, the rationale behind applying the scaling factor to the model's input strength, as opposed to its output, is not clearly articulated. This aspect raises questions, especially since category selectivity is typically associated with later stages in the visual hierarchy.

Minor:

1. In Figure 7 & 9, top panel, “normalisation” should be “normalization”.

2. On page 14, line 7, “Differences in temporal dynamics between for ventral and lateral occipital cortex” should be “Differences in temporal dynamics between ventral and lateral occipital cortex”.

3. On page 4, in the third paragraph, “retonotopic atlases” should be “retinotopic atlases”.

4. In Figure 6B and Supplementary Figure3, “AVerage ecovery (%)” should be “Average recovery (%)”

5. The full form of the abbreviation "csDN" is not provided in the manuscript.

References:

Groen, I. I., Piantoni, G., Montenegro, S., Flinker, A., Devore, S., Devinsky, O., . . . Ramsey, N. F. (2022). Temporal dynamics of neural responses in human visual cortex. Journal of Neuroscience, 42(40), 7562-7580.

Horiguchi, H., Nakadomari, S., Misaki, M., & Wandell, B. A. (2009). Two temporal channels in human V1 identified using fMRI. Neuroimage, 47(1), 273-280. doi:https://doi.org/10.1016/j.neuroimage.2009.03.078

Stigliani, A., Jeska, B., & Grill-Spector, K. (2017). Encoding model of temporal processing in human visual cortex. Proceedings of the National Academy of Sciences, 114(51), E11047-E11056.

Stigliani, A., Jeska, B., & Grill-Spector, K. (2019). Differential sustained and transient temporal processing across visual streams. PLoS Computational Biology, 15(5), e1007011. doi:10.1371/journal.pcbi.1007011

Zhou, J., Benson, N. C., Kay, K., & Winawer, J. (2019). Predicting neuronal dynamics with a delayed gain control model. PLoS Computational Biology, 15(11), e1007484.

Reviewer #2: This interesting study shows that neural adaptation differs across human visual cortex and suggests that the underlying neural computations can be explained by a model of delayed divisive normalization.

However, I have major concerns that need to be addressed.

Other forms of normalization are viable. In particular a comparison with other alternative models of normalization is critical. Otherwise it is difficult to ascertain whether a single type of model is viable. The authors also discuss fatigue and sharpening mechanisms but do not make explicit comparisons.

It is important to discuss the results in terms of excitation/inhibition ratios and principles of organization of the visual cortex.

Levels of parvocellular and magnocellular input are critical for interpretation of adaptation mechanisms and they are not addressed. The same holds true for dorsal-ventral asymmetries in adaptation. The authors could consider hysteresis signatures as a way to address adaptation mechanisms.

The level of precision of identification of visual areas is quite low and could be improved.

Stimulus durations are very short (maximum 533 ms) thereby excluding the study of adaptation of other relevant temporal scales.

The authors also ignore at this point important single cell studies on adaptation

Biological plausibility of parameter differences across the visual hierarchy is not adequately justified. The model differences in temporal normalization dynamics have little biological justification. This needs to be improved

Increased temporal windows across the cortical hierarchy may indeed have several computational benefits but this is not necessarily true in some pathways.

Reviewer #3: Brands and colleagues showed natural stimuli of varying duration, as well as two stimuli with varying inter-stimulus interval (ISI), to study the effect of adaptation and repetition suppression (RS) in patients with iEEG electrodes in early and late visual areas. They found that higher areas have slower and more pronounced adaptation to single stimuli, as well as slower recovery in the RS paradigm. They used a delayed divisive normalization (DN) model to explain their finding.

I think the question is well motivated and their experimental paradigm and analyses to be thorough overall (except a few issues as detailed below). However, I was also not surprised by these results, and therefore unsure about the novelty and significance of these results in the broader context. My main issues are as follows:

1. Since higher visual areas have higher latency to visual stimulation, and these natural stimuli are expected to produce variable responses in these areas that are involved in feature integration etc, it is not surprising that the DN model with a large delay parameter explains these results (Figure 4).

2. I feel the more nuanced results related to how normalization acts were not clearly visible in the raw data and could be due to the scaling of responses to 1. Most striking case is Figure 8A, where the preferred and non-preferred (blue and red traces) conditions are completely overlapping for neural data (top row) but show systematic differences in the DN model (bottom row). Indeed, this scaling could be an issue. If the actual response is weak beyond the noise floor, the model may try to fit it in a different way, as suggested by the authors in Figure 9. But this noise floor in the iEEG signal could be due to factors that are independent of the normalization pool. I think it would be good to show non normalized responses in some cases to make this point more explicit.

3. Likewise, the promised stronger recovery from adaptation for lower areas as shown in Figure 5B (yellow versus red) is not visible in the raw traces in Figure 5A, where both red and yellow reach almost the same level for the second stimulus for the ISI of 533 ms.

4. The number of subjects in each visual category should be mentioned upfront in the Results. It appears that majority of data comes from two subjects only.

Minor: Figure 6B y-axis should read Average recovery. ‘r’ is missing.

**Have the authors made all data and (if applicable) computational code underlying the findings in their manuscript fully available?**

The PLOS Data policy requires authors to make all data and code underlying the findings described in their manuscript fully available without restriction, with rare exception (please refer to the Data Availability Statement in the manuscript PDF file). The data and code should be provided as part of the manuscript or its supporting information, or deposited to a public repository. For example, in addition to summary statistics, the data points behind means, medians and variance measures should be available.

---

## [Decision Letter · Decision Letter 1]

28 Apr 2024

Dear Miss Brands,

Thank you very much for submitting your manuscript "Temporal dynamics of short-term neural adaptation across human visual cortex" for consideration at PLOS Computational Biology.

The reviewers acknowledged the thoroughness of your replies to their comments, but reviewer 1 still has one major comment. Based on the reviews, we are likely to accept this manuscript for publication, providing that you modify the manuscript according to the recommendation of reviewer 1 on electrode distribution.

Sincerely,

Hugues Berry

Academic Editor

PLOS Computational Biology

Daniele Marinazzo

Section Editor

PLOS Computational Biology

Reviewer's Responses to Questions

**Comments to the Authors:**

Reviewer #1: the review is uploaded as an attachment

Reviewer #3: The authors have addressed my concerns.

**Have the authors made all data and (if applicable) computational code underlying the findings in their manuscript fully available?**

Reviewer #1: None

Reviewer #3: Yes

PLOS authors have the option to publish the peer review history of their article (what does this mean?). If published, this will include your full peer review and any attached files.

Reviewer #1: No

Reviewer #3: No

Figure Files:

Data Requirements:

Reproducibility:

References:

---

## [Editor Report · Decision Letter 2]

12 May 2024

Dear Miss Brands,

We are pleased to inform you that your manuscript 'Temporal dynamics of short-term neural adaptation across human visual cortex' has been provisionally accepted for publication in PLOS Computational Biology.

Best regards,

Hugues Berry

Academic Editor

PLOS Computational Biology

Daniele Marinazzo

Section Editor

PLOS Computational Biology

---

## [Editor Report · Acceptance letter]

27 May 2024

PCOMPBIOL-D-23-01549R2 

Temporal dynamics of short-term neural adaptation across human visual cortex

Dear Dr Brands,

I am pleased to inform you that your manuscript has been formally accepted for publication in PLOS Computational Biology. Your manuscript is now with our production department and you will be notified of the publication date in due course.

With kind regards,

Zsofia Freund
